# Contextualizing time-series data: Quantification of short-term regional variability in the San Pedro Channel using high-resolution *in situ* glider data

Elizabeth N. Teel[1], Xiao Liu[1,2], Bridget N. Seegers[1,3,4], Matthew A. Ragan[1], William Z. Haskell[1,5], Burton
H. Jones[1,6], and Naomi M. Levine[1]

[1]University of Southern California, Los Angeles, California
[2]now at Princeton University, Princeton, New Jersey
[3]now at Goddard Space Flight Center, Greenbelt, Maryland
[4]now at Universities Space Research Association (USRA), Columbia, MD
[5]now at Marine Science Institute, University of California Santa Barbara, Santa Barbara, California
[6]now at King Abdullah University of Science and Technology, Thuwal, Saudi Arabia

*Correspondence to*: Naomi M. Levine (n.levine@usc.edu)

**Abstract**

Oceanic time-series have been instrumental in providing an understanding of biological, physical, and chemical dynamics in
the oceans and how these processes change over time. However, the extrapolation of these results to larger oceanographic
regions requires an understanding and characterization of local versus regional drivers of variability. Here we use high-
frequency spatial and temporal glider data to quantify variability at the coastal San Pedro Ocean Time-series (SPOT) site in
the San Pedro Channel (SPC) and provide insight into the underlying oceanographic dynamics for the site. The dataset could
be described by a combination of four water column profile types that typified active upwelling, a surface bloom, warm-
stratified-low-nutrient conditions, and a subsurface chlorophyll maximum. On weekly timescales, the SPOT station was on
average representative of 64% of profiles taken within the SPC. In general, shifts in water column profile characteristics at
SPOT were also observed across the entire channel. On average, waters across the SPC were most similar to offshore profiles
suggesting that SPOT time-series data would be more impacted by regional changes in circulation than local, coastal events.
These results indicate that high-resolution *in situ* glider deployments can be used to quantify major modes of variability and
provide context for interpreting time-series data, allowing for broader application of these datasets and greater integration into
modeling efforts.

## 1. Introduction

Time-series sites have been invaluable for providing new insights into the biological, chemical, physical, and ecological processes that occur in the world's oceans. These sites provide *in situ* measurements that are critical for model development, validation, and ongoing improvement (Fasham et al., 1990;Doney et al., 1996;Spitz et al., 2001;Boyd and Doney, 2002;Moore et al., 2002;Dugdale et al., 2002;Doney et al., 2009). To date, a small number of open ocean time-series sites (e.g. Hawaii Ocean Time-series and Bermuda Atlantic Time-series Study) have been heavily utilized by the oceanographic community while coastal time-series (e.g. San Pedro Ocean Time-series), which are more cost effective to run and so more numerous, have typically been under-utilized. One primary reason for this discrepancy is that the representativeness of these coastal sites to larger regions has been unclear. Characterizing fine-scale temporal and spatial variability at an individual site relative to a larger region is the first step in being able to leverage data from local time-series sites to gain an understanding of larger scale oceanographic dynamics.

Since most ship-based time-series are sampled at a single fixed location approximately once per month, the overall dataset is assumed to represent the mean state of a given geographical region as it varies with seasonal and annual cycles. To determine the accuracy of this assumption and to allow for the extrapolation of coastal time-series data to a larger region, high-resolution spatial and temporal monitoring of the physical and biological variability around these time-series sites is required. Because of the limitations with traditional *in situ* approaches, satellite imagery is frequently used to characterize spatial and temporal variability, assuming a tight coupling between surface and sub-surface variability (e.g., DiGiacomo and Holt, 2001;Kahru et al., 2009;Nezlin et al., 2012). However, for many coastal regions satellite observations may be insufficient for assessing the biological and environmental variability due to decoupling between surface and sub-surface dynamics, the importance of fine spatial scale (<1 km) variability, the presence of terrestrially derived chromophoric dissolved organic matter (CDOM), and cloud contamination.

High-frequency *in situ* sampling with gliders can provide uninterrupted monitoring of the surface 100 to 1000 meters vertically over tens of kilometers horizontally. These datasets provide both an understanding of kilometer-scale spatial and sub-monthly temporal dynamics as well as insight into the covariance between surface and subsurface dynamics, thereby aiding in the interpretation of satellite data. Here, we use an eight-month Slocum electric glider dataset from the San Pedro Channel (SPC) to investigate the representativeness of the coastal San Pedro Ocean Time-series (SPOT) site for the region. We demonstrate that high frequency sampling can be used to generate a framework for understanding and quantifying spatial and temporal variability in a region and to gain a better understanding of the representative nature of a given time-series location.

The SPOT station, which is located at 33° 33' 00" N and 118° 24' 00" W, sits in the SPC between Catalina Island and the Palos Verdes Peninsula where the water depth is approximately 900 meters (Figure 1). The SPC lies within the larger

Southern California Bight (SCB), which extends from Point Conception to Mexico (Noble et al., 2009b). The Channel Islands and submarine canyons oceanographically define the SCB. Within the SCB, the Southern California Eddy is a dominant, persistent feature that generates poleward-flowing surface currents that break off from the California Current (Oey, 1999; Noble et al., 2009b; Dong et al., 2009). The SCB is characterized by strong seasonal variation, including a spring upwelling season and subsequent phytoplankton blooms. Within the SPC, however, local upwelling and post-upwelling bloom formation are less persistent or predictable than observed farther north. Post-upwelling blooms occur on the timescales of days to a couple of weeks and can be quickly followed by periods of very low surface chlorophyll (Supplemental Figure S1). The SPOT station has been sampled monthly for environmental and biological parameters since 1998. Microbial communities at SPOT have been found to be annually and seasonally predictable with reoccurring phylogenetic assemblages (Fuhrman et al., 2006;Steele et al., 2011;Chow et al., 2013;Chow et al., 2014). Daily sampling at this site has shown that dominant microbial taxa vary on much shorter time-scales, indicating that monthly sampling may only represent a persistent background community (Needham et al., 2013;Needham and Fuhrman, 2016). Previous work has postulated that SPOT and the San Pedro Channel is in general representative of the larger Southern California Bight based on local circulation patterns (e.g., Cullen and Eppley, 1981;Collins et al., 2011;Chow et al., 2013). However, the ability of monthly sampling at the SPOT site to capture variability within the SPC has not been quantified. Here we present a framework for using high-resolution glider data to quantify the main modes of variability in the channel and determine whether coarse-resolution (e.g. monthly) sampling at a single point location is sufficient for understanding oceanographic dynamics within a larger region. We use the SPOT time-series site as the point location and the San Pedro Channel as the larger region. We highlight how this approach can be applied to other datasets to generate new insight into the primary modes of variability.

## 2. Methods

### 2.1 Glider Deployments

The physical and biological characteristics of the SPC were characterized using a Teledyne-Webb G1 Slocum electric glider that was deployed from March through July of 2013 and 2014. The deployment period was selected in order to maximize the likelihood that both coastal and offshore processes would be captured in the dataset (Hayward and Venrick, 1998; Di Lorenzo, 2003; Mantyla et al., 2008; Schnetzer et al., 2013). The glider was deployed on a 28 km cross-channel path between Catalina Island and the Palos Verdes Peninsula (Figure 1) and completed a single cross-channel pass every 1.5-2 days (average speed 1 km hr$^{-1}$). Data were collected between ~3 and 90 meters, with the exception of when the glider crossed the major shipping lanes where the glider was constrained to depths below 20 meters to avoid damage or loss from ship traffic. Chlorophyll-a fluorescence was measured by a WetLabs EcoPuck FL3 fluorometer, backscatter at wavelengths of 532, 660, and 880 nanometers was measured by a WetLabs EcoPuck BB3 sensor, and temperature, salinity, and pressure were measured with a SeaBird flow-through CTD. Vertical resolutions for the WetLabs pucks were approximately 0.3 m, while the vertical

resolution for the SeaBird CTD was approximately 0.6 m. The glider was recovered every 3-4 weeks for cleaning, battery replacement, and recalibration using standard methods published in Cetinić et al (2009).

**2.2 Ancillary Satellite Data**

Level 3 mapped MODIS Aqua daily 9km photosynthetically active radiation (PAR) measurements were acquired from the NASA Ocean Biology (https://oceandata.sci.gsfc.nasa.gov/MODIS-Aqua/Mapped/Daily/9km/par/). MODIS Aqua daily 1 km chlorophyll (ChlSat) data were acquired from NOAA CoastWatch West Coast Regional Node (http://coastwatch.pfeg.noaa.gov/coastwatch/CWBrowser.jsp). These data were then matched geographically and temporally with the *in situ* glider data.

**2.3 Glider Data**

Glider data were processed, calibrated, and quality controlled following Cetinić et al. (2009) and Seegers et al (2015). To correct for current induced drift, the glider data from each 2-day transect were gridded onto an idealized glider transect with 500 m horizontal resolution and 1m vertical resolution that was approximately perpendicular to the mean flow and the coastline (Figure 1). Only glider data within 5 km of the idealized transect were used in this analysis (Figure 1). Each 500 m bin (N=62) corresponded approximately with a single downcast and upcast. Only profiles with data for >85% of the vertical bins were used for further analyses, thereby excluding partial profiles from under the shipping lanes. The remaining missing data (< 15% of each profile) were filled using 2D interpolation from all neighboring bins. A total of 557 profiles from 2013 and 1049 profiles from 2014 were accepted for further analyses. Of the 1606 final glider profiles, 1151 matching PAR measurements and 571 matching ChlSat measurements were available.

For each profile, the mixed layer depth (MLD) was calculated as the depth where the density change exceeded the equivalent of a 0.4˚C temperature drop relative to the density at 5 m (modified from Sprintall and Tomczak, (1992)). The mixed layer temperature (MLTemp) was calculated as the mean temperature within the mixed layer. The light field between 1 and 80 m was calculated for each glider profile following the regionally validated method described in Jacox et al. (2015). This method uses surface PAR measurements and *in situ* chlorophyll a profiles to calculate the diffuse attenuation coefficient at each depth. The euphotic depth, defined as the 1% light level, was then calculated for each glider profile from these light profiles (Kirk, 1994). The glider based euphotic depths were in good agreement with those collected from *in situ* PAR measurements during the concurrent Upwelling Regime In-Situ Ecosystem Efficiency study (UpRISEE) cruises at the SPOT site (Haskell et al., 2016). We also calculated the first optical depth (OD1) for each glider profile as the depth in meters where available PAR was equal to 1/e of surface PAR after Gordon (1975) and Kirk (1994).

The temperature, salinity, and chlorophyll a data from each of the 1606 glider profiles were used to calculate secondary metrics that were used for statistical analyses. Specifically, maximum chlorophyll fluorescence (MaxCHL), depth

of maximum chlorophyll fluorescence (zMaxChl), 70 meter integrated chlorophyll (ChlInt70), depth of maximum backscatter (zMaxBB), maximum backscatter (MaxBB), and ratio of integrated chlorophyll in the top 70 meters relative to the integrated chlorophyll in the top 20 meters (ChlInt70Per20) were calculated. Twenty meters was used to approximate the average mixed layer depth. Seventy meters was chosen as the maximum depth of chlorophyll integration as it included the full euphotic depth for 99% of the glider profiles from 2013 and 2014. In addition, we estimated from the ship-based SPOT time-series data (2003 - 2011) that on average PAR at 70m was 2.6% of the surface value, with a maximum of 4.5%. Finally, the depth of the 12.5˚C isotherm (z12p5) was used as a proxy for the top of the nutricline, which indicated nutrient-rich sub-thermocline waters in the SPC and within the CalCOFI region (Hayward and Venrick, 1998; Lucas et al., 2011). This relationship was confirmed for the SPOT site using ship-based nitrate and temperature data.

Depth profiles of primary production (PP(z)) were estimated after Jacox et al. (2015) as:

$$PP(z) = pB(z) * chl(z) * dirr$$

where chl(z) is the chlorophyll concentration (mg chl m$^{-3}$) from the glider profiles, and dirr is day length (hrs day$^{-1}$). pB(z) is a regionally-tuned and light-dependent carbon fixation rate in mg C mg chl$^{-1}$ hr$^{-1}$ that was calculated from PAR profiles as per Jacox et al. (2015). Integrated primary production over the euphotic depth and the first optical depth (OD1) were calculated for each glider profile. Integrated primary production estimates for SPOT were within the bounds of regional estimates (Jacox et al., 2015).

## 2.4 Principal Component Analysis (PCA)

Principal component analysis (PCA) was used to differentiate between the major water column profile types observed within the glider dataset. All glider profiles from 2013 and 2014 were combined into a single PCA after normalization and standardization of the profile characteristics as described above. A step-wise PCA was conducted to determine the relative influence of each of the secondary characteristics on total observed variance within the end-member profile dataset. Based on this analysis, two characteristics (zMaxBB and MaxBB) were omitted from further PCA analyses as they did not strongly affect overall dataset variance or the resulting PCA distribution. In order to generate a framework for analyzing water mass variation at the site, results from the original PCA were used to select a subset of profiles that described 'end-member' water column profile types (*details of this selection are described in section 3.2*). A second PCA was then conducted using the subset of glider profiles (N=54) and the same set of secondary characteristics (referred to as a structured PCA). The remaining glider profiles (N=1552) were projected onto the structured PCA axes using the function proj within R software. Confidence intervals of 95% were calculated for each clustering in PCA space using the iso-contour of the Gaussian distribution after www.visiondummy.com/2014/04/draw-error-ellipse-representing-covariance-matrix/ and the function ggbiplot within R software. In brief, the magnitude of ellipse axes were determined by the variance within each cluster, defined as the eigenvalues from the covariance matrix. The direction of the major axis was calculated from the eigenvector of the covariance matrix that

corresponded to the largest eigenvalue. The loadings of the secondary characteristics onto the PCA axes were also calculated and plotted.

## 2.5 Comparison with time-series measurements

To interpret the San Pedro Ocean Time-series data (monthly sampling) within the context of the variability identified in the high-resolution glider dataset, we incorporated 12 years of available ship-based measurements at the site into our analysis. Specifically, profile characteristics (described above) were calculated for 64 ship-based SPOT profiles from 2000-2011. Thirty of these profiles fell between March and July, the months during which the gliders were in the water. An additional 21 profiles were calculated from the UpRISEE ship-based cruises that occurred every two weeks during 2013 and 2014 (Haskell et al., 2016), with 14 profiles occurring between March and July. Though there was good coherence between temperature measurements across all three datasets, the chlorophyll fluorescence measurements from the 2000-2011 SPOT site cruises were considerably lower than the fluorescence measurements from both the in situ gliders and the ship-based UpRISEE cruises from 2013 to 2014. We assumed this to be inter-instrument variation in fluorescence to chlorophyll ratio, rather than changes in in situ chlorophyll concentration itself. To allow for projection onto the glider-derived structured PCA axes, the chlorophyll fluorescence data from 2000-2011 SPOT cruises were scaled so that the March through July mean chlorophyll fluorescence value was equal to the March through July mean chlorophyll fluorescence value from the 2013 to 2014 UpRISEE cruises. As all SPOT chlorophyll a data was scaled together, this correction will not impact the relative distances between samples in PCA space but was necessary to allow for comparison to the glider data. The profile characteristics (MLD, MLTemp, z12p5, zMaxChl, maxCHL, chlInt70, and chlInt70Per20) for the 85 ship-based profiles were used to project these samples onto the structured PCA axes. Corresponding PC1 and PC2 values for all ship-based SPOT site profiles were then compared with glider profiles to assess interannual profile variability at the SPOT site.

## 3 Results

### 3.1 Cross-Channel Oceanographic Trends

Cross-channel comparisons of mixed layer temperature (MLTemp), mixed layer depth (MLD), depth of the 12.5°C isotherm (z12p5), integrated chlorophyll over the upper 70 m (ChlInt70), and integrated primary production within the euphotic zone were used to identify persistent oceanographic gradients across the transect (Figure 2). Observed physical properties in both spring 2013 and spring 2014 displayed an onshore-offshore gradient, where the onshore direction was defined as towards the Palos Verdes Peninsula (PV) and offshore was defined as towards Catalina Island. This gradient could be seen most clearly in the z12p5 data, where there was a strong offshore tilt in the mean depth of this isotherm (Figure 2c). This tilt is consistent with equatorward flow through the channel, which frequently occurs during the spring (Hickey et al.,

2003;Noble et al., 2002), but could also have been amplified closest to shore with periods of active upwelling. Without sampling the surface data within the shipping lanes, it was not possible to determine the full profile behaviour of locations between the coastal and mid-channel bins. However, subsurface glider temperature data showed that weakened stratification during early upwelling events often extended across the entire SPC (Supplemental Figure S4). These observations suggest that there is coherence across the shipping lanes during upwelling events.

Though the average depth of the cold, high-nutrient waters was shallowest close to shore, the cross-channel data for ChlInt70 displayed only a weak cross-channel gradient (Figure 2d). Rather, ChlInt70 had fairly constant cross-channel values of about 100 mg chl m$^{-2}$. It is important to note that integrated chlorophyll alone cannot be used to assess the productivity of a location because it does not account for the vertical chlorophyll distribution and its overlap with the vertical light field. In fact, depth integrated primary production displayed a much stronger an onshore-offshore gradient in both mean and variance than was observed for ChlInt70. As expected, the highest primary productivity was observed closest to the mainland with decreasing values observed across the channel away from coastal upwelling sites (Figure 2e).

To compare cross-channel differences in water column profile characteristics, three bins were selected: bin 10 (near Catalina, n = 41 profiles), bin 28 (SPOT, n = 54 profiles), and bin 58 (near the mainland, n= 34 profiles). When the variance at each location was taken into account, the mixed layer depth (MLD) and integrated primary production (PP) for bins 10 and 28 were not significantly different from one another (two-sample $t$-test, p-value < 0.01) while bins 10 and 28 were both significantly different from bin 58 (two-sample $t$-test, $p \ll 0.01$). Mixed layer temperature and integrated chlorophyll (ChlInt70) were not significantly different across all three bins. The mean depth of the 12.5° isotherm (z12p5) was significantly different for all three bins (two-sample $t$-test, $p \ll 0.01$). These cross-channel analyses highlight high intra-bin temporal variability over the course of the deployments, especially for ChlInt70 and MLD (Figure 2). The importance of this temporal and spatial pattern of variability for the SPC is twofold: 1) a monthly time-series sampling scheme at SPOT may under-sample both biological and physical variability in the channel, however 2) given the similarity in variance across the SPC, with sufficient sampling, SPOT data could be representative of the average state of the SPC.

### 3.2 Dominant water column profile types

To identify dominant water column profile types within the SPC, the glider profiles were analyzed using an iterative principle component analysis (PCA). First, a standard PCA was conducted using all glider profiles to identify overarching patterns within the data. This original PCA was used to identify four end-member profile types that were then used to generate a second PCA (structured PCA). The structured PCA provided clear separation between the end-member profile types and provided a useful framework for analyzing seasonal and interannual variability within the dataset.

The original PCA (all profiles) showed that most profiles were clustered together, with a small subset of cold, high chlorophyll (nominally surface bloom) profiles driving much of the separation on both the first principal component (PC1) and second principal component (PC2) (Figure 3a). Analysis of the PCA suggested that there were significant environmental

and ecological differences (e.g. temperature, stratification, chlorophyll content) between profiles within the large cluster. Specifically, two end-members within this cluster were apparent: a cool, deep MLD, low chlorophyll water column profile type and a warm, shallow MLD, low surface chlorophyll water column profile type. Based on the PCA, we defined three 'end-member' water column profile types: (1) cool, high chlorophyll (CHC); (2) cool, low chlorophyll (CLC); (3) warm, subsurface high chlorophyll (WSHC). In addition, we identified a fourth, unique, end-member water column profile type based on our examination of the glider dataset and our understanding of the oceanography of the San Pedro Channel. This fourth type represented an oligotrophic end-member with a warm, shallow MLD, and low chlorophyll throughout the water column and was termed: (4) warm, low chlorophyll (WLC).

These four water column profile types are consistent with our understanding of oceanographic states of the region. Specifically, there are two primary physical dynamics that impact water column signatures in the SPC (Hickey, 1979; Kim et al., 2014; Noble et al., 2009b; Dong et al., 2009): 1) coastal upwelling and 2) the Southern California Eddy. Periodic coastal upwelling in the spring brings cool, high nutrient waters to the surface and triggers large surface blooms that extend from the coast into the channel. The cool, low chlorophyll (CLC) and cool, high chlorophyll (CHC) end-members represent the beginning and end of this process. In the late spring and early summer, seasonal heating and the spin-up of the Southern California Eddy bring warm low nutrient waters into the SPC. These waters have a distinct signature that we identify here as warm, low chlorophyll (WLC) waters. The mechanism generating the fourth end-member profile type – warm, subsurface high chlorophyll (WSHC) – is less clear. There are two leading hypotheses: 1) these are either coastal surface blooms and/or elevated nitrate concentrations from coastal upwelling that have been advected along sub-surface isopycnals out into the channel (e.g., Mitarai et al., 2009; Bialonski et al., 2016; Stukel et al., 2018) or 2) that internal waves result in isopycnal heave of nutrients into the euphotic zone creating enhanced chlorophyll concentrations (e.g., Noble et al., 2009a; Noble et al., 2009b; Lucas et al., 2011).

All instances of these four end-member profile types were identified in the full data using MLTemp, ChlInt70Per20, ChlInt70, z12p5, maxCHL, zMaxChl, and MLD criteria (Supplemental Table S1). For end-member types 1-3, we started with the profiles identified in the original PCA and refined the criteria in order to isolate the most 'pure' examples of these water mass profile types. Using our criteria, we identified 54 'end-member profiles': 10 for type 1 (CHC), 12 for type 2 (CLC), 15 for type 3 (WSHC), and 17 for type 4 (WLC). Average temperature and chlorophyll profiles for all four end-member types are shown in Figure 4. CLC profiles were characterized by MLTemps that were cooler than 12.5˚C, a shallow chlorophyll maxima, and integrated chlorophyll that did not exceed 85 mg chl m$^{-2}$. The combination of these characteristics indicated that the deep, cold, nutrient-rich water had been recently advected into the surface mixed layer. CHC profiles were characterized by MLTemps between 13°C and 17°C, MLDs deeper than 10 m, and ChlInt70 values above 150 mg chl m$^{-2}$, indicating that upwelling had begun to relax, the 12.5 ˚C isotherm had returned to a depth of greater than 20 m, and a strong surface bloom was present. To clearly differentiate surface blooms from subsurface chlorophyll maxima, the integrated chlorophyll within the surface 20 m was directly compared to that within the top 70 m. Here we define surface blooms as those with at least 50% of the total integrated chlorophyll within the top 20 m. Both WLC and WSHC profiles were characterized by high MLTemp,

shallow MLD, and deep z12p5. WLC profiles were relatively oligotrophic and had values of ChlInt70 less than 80 mg chl m$^{-2}$ while WSHC profiles had values of ChlInt70 up to 142 mg chl m$^{-2}$ but with less than 5% of integrated chlorophyll in the top 20 m of the profile.

To more efficiently differentiate profile characteristics and to better represent the observed variability within the glider profile types, we conducted a second PCA, which used only the 54 'end-member' profiles to define the PCA axes (hereafter referred to as the structured PCA). The structured PCA resulted in clear separation of the 4 end-member profile types and allowed for better overall separation of the glider profiles (Figure 3b). Specifically, the percent of explained variance for PC1 increased from 40.5% to 49.8%, and the percent of explained variance for PC2 increased from 21.5% to 32.7%. More importantly, the structured PCA allowed for more meaningful separation of the glider profiles into oceanographically relevant states allowing us to better understand and quantify the overall variance within the glider profile data. Movement along PC1 primarily described changes in temperature-based characteristics while movement along PC2 primarily described changes in chlorophyll-based characteristics (Figures 3b and 4; Supplemental Figure S2). Specifically, PC1 was most positive when MLTemp was greater than 19 ˚C and the seasonal thermocline was strongly defined. PC2 was most negative when the chlorophyll maximum was most clearly defined and ChlInt70 was greater than 120 mg chl m$^{-2}$. To determine the relative contribution of the 4 water column profile types to the observed glider profiles, the 1606 final glider profiles were projected onto these end-member axes (Figure 3b). As PC1 and PC2 explained similar amounts of variance within this dataset, smaller 'distances' between profiles in PC coordinate space approximates similarity in profile features. As the four 'end-member' water column profile types were selected to represent the end-member profile characteristics, one would anticipate that a monthly time-series would, for the most part, capture intermediate states rather than these end members. Identifying these end-members both helps characterize the overall biological and physical variation seen in water column profiles within the SPC and provides a means for quantifying the influence of coastal (CLC, CHC) versus offshore (WLC, WSHC) waters in the channel.

## 3.3 Variability at the SPOT Station

The glider profiles were analyzed within the structured PCA space with specific focus on temporal and spatial changes, as well as how the profiles taken at the SPOT site related to and varied with profiles from the rest of the San Pedro Channel cross-section. When all glider profiles were projected onto the PCA axes, 39.6% fell within one of the end-member profile type clusters, as determined by 95% confidence intervals (Table 1). Similarly, 37% of the 54 SPOT glider profiles were associated with an end-member subgroup. Of the four end-member profiles, the offshore-types (WLC, WSHC) were most prevalent with 24.8% of all glider profiles falling within the WSHC cluster and 12.1% aligning with WLC cluster. SPOT profiles followed a very similar trend with 25.9% of profiles falling within the WSHC cluster and 9.3% of profiles aligning with the WLC cluster. The two coastal end members (CLC, CHC) were rare in the overall dataset as well as in the SPOT profiles themselves with less than 4% of all profiles associated with these coastal end-members. These results indicate that,

for the majority of locations and times sampled during the spring and early summer of 2013 and 2014, water column profiles within the SPC most closely resembled offshore profiles. These results most likely underrepresent the overall coastal influence within the SPC due to the exclusion of data from within the shipping lanes that influenced 30% of the transect and were located close to the coast (Figure 1). However, these findings are consistent with previous research conducted at SPOT that indicated a general prevalence of open ocean bacterial groups (Chow et al., 2013) and a notable scarcity of land-derived organic matter (Collins et al., 2011). It is important to note that since the general circulation in the SPC is parallel to shore and perpendicular to the glider transect, many signatures observed in the dataset may be upstream processes that were advected into the study domain. While our end-member framework allows us to quantify the occurrence of upstream coastal signatures within the dataset and variability in water mass characteristics, we cannot distinguish between local and remote sources of these water masses. For example, we cannot determine if CLC waters upwelled within the SPC or whether they were upwelled near the Santa Barbara Channel or Point Dume and advected into the San Pedro Channel. To improve detection of upstream influence within the SPC in future deployments, the glider transect would need to be adapted to incorporate alongshore movement as well as cross-channel travel.

To further investigate how changes in the SPOT profile characteristics were related to co-occurring changes in profile types across the channel, we looked at the relationship between all non-SPOT profiles and the corresponding SPOT profile. For this analysis, the PCA coordinate plane was divided into four quadrants representing low biomass and coastal dynamics (positive PC2, negative PC1), low biomass and offshore dynamics (positive PC2, positive PC1), high biomass and offshore dynamics (negative PC2, positive PC1), and high biomass with coastal dynamics (negative PC2, negative PC1) (Figure 5). These four quadrants were then used to investigate the cross-channel distribution relative to the most recent SPOT profile. The majority of profiles (64%) fell into the same PCA quadrant as the most recent SPOT profile suggesting that the majority of the channel displayed similar profile characteristics and that the SPOT site in general captured across channel variability. Broken down by profile type, SPOT profiles were most similar to co-occurring cross-channel profiles when SPOT displayed WSHC profile characteristics (78% of cross-channel profiles in the same quadrant as the SPOT profile, Figure 5d). SPOT profile characteristics coincided with 55% of other profiles when exhibiting characteristics of WLC, 48% when displaying CHC characteristics, and 33% of the channel in the one instance when the SPOT profile displayed more coastal physical CLC properties (Figure 5). The sample distributions in each of these four cases were statistically different from one another (two-tailed $t$-test, $p<0.01$), indicating that the profile characteristics at the SPOT site were indicative of the state of the SPC as a whole. For example, when SPOT profiles showed the most coastal signatures (negative PC1) (Figure 5a & 5c), no channel profiles were found to be representative of either the WLC or WSHC profile types. Conversely, during periods when SPOT profiles were most similar to offshore types (positive PC1) (Figure 5b & 5d), no channel profiles were found to be representative of CHC profile types.

Due to seasonal variations in local upwelling and the spin-up of the Southern California Eddy, temporal dynamics and succession play an important role in driving regional productivity. To track the seasonal progression of water mass profile types for SPOT relative to more near-shore bins, the time-series of profile characteristics for SPOT (bin 28) and coastal bins

(55-62) were analysed in PCA space (Figure 6, Supplemental Figure S3). In April, both the nearshore and SPOT bins showed significant coastal influence with profile types with CLC and CHC signatures. During late April to early May of each deployment, the nearshore and SPOT bins collectively experienced a relative 'cross-over', when the profiles shifted from some coastal influence to offshore dominated characteristics. In June, both the nearshore and SPOT bins showed offshore influence with profile types with WLC and WSHC signatures. These results suggest cross-channel covariance and that profile variability

in both mid-channel and nearshore locations was likely forced by the same oceanographic dynamics.

### 3.4 Surface to Subsurface Coherence

    Our analyses indicate that subsurface chlorophyll maximum characteristics were a dominant profile type within the SPC (occurring ~25% of the time), and that these subsurface chlorophyll maxima (WSHC) may appear similar to offshore water intrusion (WLC) from surface characteristics alone. In this study, the subsurface chlorophyll maximum was on average

only 5 m deeper than the particle maximum, determined using the back-scatter maximum. The subsurface particle maxima observed within this study were consistent with previous regional findings (Cullen and Eppley, 1981). The close proximity between the particle and chlorophyll maxima suggests that these subsurface phytoplankton communities may contribute significantly to local primary production (Cullen and Eppley, 1981). While satellite surface chlorophyll estimates have been previously shown to align closely with *in situ* glider observations of nearshore surface blooms in the San Pedro Channel,

subsurface chlorophyll layers farther offshore were undetected by satellite retrievals (Seegers et al., 2015). Here, we used our framework to identify which oceanographic states for the SPOT site may be most susceptible to satellite misinterpretation.

    The first optical depth (OD1) was calculated from satellite PAR and glider chlorophyll-based diffuse attenuation coefficients for PAR (Kirk, 1994; Jacox et al., 2015). This method provides a conservative estimate of OD1 by ignoring light attenuation caused by dissolved organic matter. Average OD1 for these deployments was 12.3 m while the average euphotic

depth, as defined by the 1% light level, was 38.3 m. This is consistent with *in situ* measurements of the euphotic depth from temporally overlapping cruises at the SPOT site that observed an average euphotic depth of 40 m (Haskell et al., 2016). Our estimates for the first optical and euphotic depths are also within the range of regional data collected during the CCE LTER process cruises from 2006-2008 (Mitchell, 2014). To investigate the potential bias in quantifying chlorophyll only over OD1, we compare glider estimates of chlorophyll and primary production integrated over the entire euphotic depth to glider estimates

integrated over the first optical depth. Using glider data analyzed over different depth intervals provides an internally consistent dataset and eliminates other confounding factors that might impact the analysis. MODIS Aqua *chlorophyll a* data was also acquired and compared with the *in situ* glider data. However, a low correlation between glider and satellite integrated chlorophyll over OD1 was observed, which could be due to temporal and spatial mismatches between datasets, inaccurate CDOM corrections, poor atmospheric correction, and high subsurface biomass (Supplemental Figure S5).

Overall 87% of integrated chlorophyll within the euphotic zone was located beneath the OD1 during the 2013 and 2014 deployments. This percentage increased to 92% for samples with a significant subsurface chlorophyll maxima (i.e. WSHC)

and decreased to 82% for surface blooms (i.e. CHC). For these deployments, zChlMax was generally at or below the 10% light level, deepening with more oligotrophic conditions. As a result of these differences in subsurface chlorophyll profiles, integrated primary production over OD1 varied from 1.8% to 71.3% of integrated primary production over the entire euphotic zone. By analyzing glider profiles based on water column characteristics, we were able to identify profile types that may be more susceptible to the inherent bias in satellite data of only quantifying chlorophyll over the first optical depth. Specifically, for samples with CLC signatures (high PC2 values and low overall biomass), a significant correlation was observed between OD1 and total integrated chlorophyll ($r^2=0.55$, $p<0.01$) and between OD1 and total integrated primary production ($r^2=0.77$, $p<0.01$). This relationship weakened as surface biomass increased with no correlation observed between OD1 and total integrated chlorophyll for profiles falling within the CHC 95% confidence interval cluster ($p=0.62$). While, OD1 integrated primary production was still significantly correlated with total integrated primary production for CHC profiles ($r^2=0.36$, $p<0.01$), the slope of this relationship changed significantly from that observed for CLC profiles from 2.5 to 0.6 (Figure 7). Similarly, profiles with offshore characteristics (WLC, WSHC) showed a correlation between OD1 and total integrated primary production ($r^2=0.34$ and $r^2=0.31$ respectively, $p<0.01$) but with slopes of 1.6 and 1.3.

These results quantitatively illustrate differences in the relationship between surface properties (remote-sensing observable) and water column integrated properties. Moreover, we demonstrate that the relationship between OD1 integrated and euphotic zone integrated primary production shows large variations with water column profile type and that the relationship between surface and water column integrated values is more predictable for certain profile types than others. Our analysis suggests that the satellite observable vs integrated euphotic zone chlorophyll mismatch may be particularly problematic for some cool, high chlorophyll water mass types (nominally coastal blooms). This suggests that increased *in situ* sampling may be needed when these water mass types are present in order to accurately constrain estimates of biomass distributions and primary production. Oligotrophic profiles (WLC, WSHC) also present a challenge with OD1 integrated primary production only explaining approximately a third of the variability in total integrated primary production. However, the presence of a pronounced subsurface chlorophyll maximum did not substantially change the relationship between OD1 integrated and euphotic zone integrated PP as compared to oligotrophic samples with similar OD1 integrated chlorophyll but without the subsurface maximum. This analysis suggests that remote-sensing based estimates of PP for WSHC profiles would not be more biased than WLC profiles.

### 3.5 Context for time-series measurements

The PCA framework for quantifying variability within the San Pedro Channel described above can be leveraged to provide context for the SPOT time-series site measurements, which are made at a single location with monthly resolution. Figure 3b shows the projection of ship-based SPOT data from 2000-2011 and data from a set of Upwelling Regime In-Situ Ecosystem Efficiency study (UpRISEE) cruises that occurred during the time of the glider deployments (2013-2014) onto the structured PCA axes. Ship-based measurements from March-July show a similar distribution of profile types as seen in the

glider profiles with most profiles showing an offshore signature (WLC, WSHC) (Table 1, Supplemental Figure S6). There was not a substantial difference between the distribution of ship-based profiles in March-July and the distribution for the full ship-based datasets. This suggests that there is not a significant seasonal shift in variability at the SPOT site and that the variability observed in the glider dataset may be relevant for the entire time-series dataset.

The impact of sampling frequency on the observed profile characteristics at SPOT was tested by subsampling the glider dataset to evaluate the difference in ~4-day (full glider dataset, N=54 profiles), 8-day (2 datasets, N=27 profiles), 12-day (3 datasets N=18 profiles), 16-day (4 datasets N= 14 profiles), 20-day (5 datasets N=11 profiles) and 24-day (5 datasets N= 9 profiles) sampling schemes. The number of subsampled datasets varied as a result of sub-sampling a dataset of fixed length with 4-day sampling– for example only 2 unique datasets could be generated with 8-day subsampling from the original dataset with 4-day sampling. Differences in the mean and standard deviation of the low versus high-frequency sampled datasets in PCA space were used to evaluate the impact of sampling frequency on estimates of the mean state at the SPOT site. As expected, less frequent sampling resulted in more variability in the average water column profile characteristics with 8-day sampling showing the most similarity to the full dataset (4-day sampling) (Figure 8). There was not a noticeable difference between 12-day, 16-day, 20-day and 24-day sampling (Figure 8a). Overall, the differences between the 6 different sub-month sampling schemes were small compared to the observed spread in the full glider dataset (Figure 8b). This analysis indicates that monthly sampling at the SPOT site may be sufficient to capture the majority of the observed variability in the SPC due to the strong offshore influence at the site, even during the upwelling season. However, as highlighted above (*section 3.3*), monthly sampling will most likely underestimate the impact of infrequent events at the site such as strong surface blooms and upwelled waters. Similarly, a chance sampling of a rare event (e.g. upwelling or coastal bloom) might skew the estimated monthly state for SPOT given that these events appear to be short-lived at the SPOT site.

The structured PCA framework also allowed us to analyse interannual variability in water column characteristics at the SPOT site. While substantial interannual variability in profile characteristics was observed, as represented by changes in PC1 and PC2, the variability in water column profiles captured by the high-resolution gliders during the upwelling period (March-July) of 2013-2014 was comparable to the interannual and seasonal variability observed in monthly sampling at SPOT over 12 years (Figure 9). Deviations from the mean state (e.g. 2000) could indicate anomalous oceanographic conditions or chance sampling of stochastic rare events (e.g. capturing a spring bloom event). Further analysis of the SPOT data within this context could potentially tease apart these two signals.

## 4. Discussion

### 4.1 Regional Application of SPOT Data

This study identified four water column major profile types within the SPC during March through July of 2013 and 2014, cool low chlorophyll, cool high chlorophyll, warm low chlorophyll, and warm subsurface high chlorophyll. These four

major water column profile types represented end-member oceanographic states that occurred within the SPC during consecutive glider deployments as a result of two primary physical drivers: coastal upwelling and the Southern California Eddy. The statistical analysis of these profile types allowed for the identification of temporal and spatial trends in local variability within the SPC and the contextualization of the SPOT station within the SPC. Our results highlight both the similarity between the SPOT station and offshore profiles (WLC, WSHC) and the similarity in water mass characteristics across the channel. This suggests that data collected at SPOT during monthly time-series sampling should be representative of the SPC. As such, long-term ecological or physical trends identified within the SPOT time-series data may be assumed to be applicable for the SPC and could give relevant insight into offshore regions within the Southern California Eddy as well. While infrequent events such as coastal upwelling and surface blooms do occasionally extend across the channel, our findings suggest that monthly sampling at SPOT will under-sample these biogeochemical fluctuations even during the coastally-dominated early Spring which experiences the most coastal influence.

During this study, glider deployments were timed to sample only during the most dynamic months for the SPOT station. This sampling scheme may have missed some seasonal variability caused by non-upwelling driven events, such as submesoscale eddies. However, our results suggest that a year-long sampling scheme would likely find similar relationships, given that the sampling occurred during the season when coastal processes (local upwelling) were most likely to occur but still concluded that SPOT was most representative of offshore water types. This is also supported by the similarity between variability with the glider dataset and the 12-year SPOT time-series dataset, which consists of monthly measurements taken throughout the year. Building on these findings, and understanding the rarity of coastal events extending across the SPC, higher-frequency time-series sampling at SPOT during the spring season could be used to better monitor the effects of nearshore processes on the physical and biological oceanography of the SPC. However, even sub-monthly sampling at SPOT may underestimate productivity and export within the SCB, as most of the observed upwelling signature occurred inshore of SPOT.

The offshore state of SPOT during this study also suggests that time-series data collected at SPOT would be sensitive to climate-derived changes to major current patterns and wind patterns, but may be relatively insensitive to local terrestrial changes such as decreased freshwater discharge or increased sedimentation. Within the SPC, the spin-up of the Southern California Eddy is a major driver of oceanographic change. We hypothesize that the eddy is the primary driver of the profile shift during late April and early May of 2013 and 2014 observed at both SPOT and nearshore locations. Utilizing our framework to identify transitions from coastally-dominated to offshore-dominated profile types, the timing of the spin-up could be monitored historically or prospectively within ship-based sampling at the SPOT site.

**4.2 Determining Regional Domains for Time-Series Sites**

The framework developed to study the applicable domain for SPOT samples and the sensitivity of SPOT samples to regional oceanographic dynamics could easily be applied to other time-series sites. The glider deployments in this study were

designed to sample a dynamic period within the SPC with high-spatial and high-temporal frequency. These high frequency samples enabled identification and statistical analysis of cross-channel covariance and response to local oceanographic events. In the case of the SPC, these events were upwelling and the subsequent surface bloom, and the spin-up of the Southern California Eddy. While these dominant water column profile types will vary with oceanographic region, the same methodology could be applied to any region to provide context for a time-series site. For example, at an open ocean site such as the Bermuda Atlantic Time-Series site these features could include winter mixing, and cyclonic or anti-cyclonic eddies. We have shown that high-frequency data can be used to identify the major regional modes and water column profile types, which can then be used to determine long-term trends within the time-series site itself. These profile signatures allow for a more quantitative description of time-series observations and therefore a more accurate method for detecting deviations and long-term trends within the time-series. In the example of the SPOT station, increased nearshore upwelling or delayed spin-up of the Southern California Eddy would each increase the overall coastal signature of vertical profiles collected at SPOT. Historical ship-based vertical profiles as well as future profiles could be analyzed with reference to coastal and offshore signals in order to determine changes in the dominant oceanographic drivers over time.

## 5. Conclusions

*In situ* high-frequency regional data collection can contextualize time-series sites and identify the major modes of regional variation. This not only allows for the data to be more accurately extrapolated to provide larger scale estimates of critical dynamics such as primary production or carbon export, but also identifies the most crucial regional signals that would need to be correctly simulated to produce accurate regional modeling. In the case of the San Pedro Ocean Time-series (SPOT) station, this study identified four major regional water column profile types and suggests that time-series samples collected monthly would in general be most representative of offshore profiles. Glider profile data from the SPC also indicate that integrated primary productivity of surface bloom profiles may be underestimated by satellite chlorophyll measurements, suggesting that accurate observation of regional dynamics within the SPC may require *in situ* sampling with increased resolution when these profile types are present. Finally, our analysis indicates that SPOT is primarily reflective of the offshore stratified environment and only rarely influenced by near-coastal processes such as upwelling. This suggests that long-term changes in the SPOT time-series dataset are more likely to reflect larger-scale regional responses (e.g. climatic shifts) than local events (e.g. increased discharge into the port of Los Angeles).

Without context for time-series data, physical or temporal under-sampling at time-series stations may mask local drivers of variability making it difficult to accurately scale-up local results to inform larger-scale analyses and modeling efforts. The methods described in this study can be applied to other coastal and oceanic time-series sites in order to identify the major modes of local variability, the region represented by the time-series data, and the sensitivity of the site to anthropogenic change. Better understanding of the spatial domain represented by global marine time-series sites will aid in the extrapolation of local findings, in the improvement of regional modeling, and in the coupling of regional and global modeling efforts.

**Acknowledgments**

This work was funded by NOAA ECOHAB, NSF-ChemOce (1260296 to M. Prokopenko and 1260692 to D. Hammond), NSF-OCE 1323319, the University of Southern California, a USC Provost Fellowship, the NSF graduate research fellowship program, the Wrigley Institute for Environmental Sciences, and the King Abdullah University of Science and Technology. We would like to acknowledge the help of C. Oberg, N. Rollins, A. Gellene, I. Cetinic, A. Pereira, R. Arntz, K. Heller, D. Diehl, T. Gunderson, D. Kim, the Fuhrman lab, the Sukhatme lab, the Caron lab, the Southern California Coastal
Water Research Project, the crew of the Yellowfin, and many others at USC who helped with SPOT data collection and with glider deployment, recovery, and maintenance.

**Figure Captions:**

**Figure 1:** Glider deployment map and idealized glider transect. Slocum electric gliders were deployed in 2013 and 2014 between the Palos Verdes Peninsula and Catalina Island in the San Pedro Channel. The San Pedro Ocean Time-series (SPOT) station, located at 33° 33' 00" N and 118° 24' 00" W, is indicated by the red dot. Glider surfacings are indicated with grey dots. An idealized transect was defined running perpendicular to the mean flow (thick black line). Glider profiles collected within 5 kilometers of the idealized transect line (dashed black lines) were used to assess cross-channel variability of oceanographic properties during these glider deployments. The location of bin 10, 28 (closest to SPOT), and 58 are shown in red. The distance from bin 10 to Catalina Island is 5.3 km, from bin 10 to bin 28 is 9 km, from bin 28 to 58 is 15 km, and from bin 58 to the closest point on the mainland is 4.8 km. The bathymetry of the study area is designated by the color contours and ranged from ~ 20 - 900 m. The maximum glider dive depth was 90 m.

**Figure 2:** Cross-channel variation of profile characteristics. Cross-channel variation in mixed layer temperature (a), mixed layer depth (b), the depth of the 12.5°C isotherm (c), vertically integrated chlorophyll (surface to 70 meters, d), and vertically integrated primary production (surface to euphotic depth, e) for gridded glider profiles from March through July of 2013 and 2014 are shown. Cross-channel box plots show the median value for each bin (white dot), data between the 25th and 75th percentiles (black box), data between the 9th and 91st percentiles (black lines), and outliers (black circles). Low numbered bins correspond with the western side of the San Pedro Channel (SPC), near Catalina. High numbered bins correspond with the eastern side of the SPC, near the Palos Verdes Peninsula (PV). Bins 35-55 correspond with the shipping lanes for the Port of Los Angeles and so have been removed due to incomplete profiles (<85%).

**Figure 3:** Principal Component Analysis of glider profiles. The original PCA (a) and structured PCA (b) are shown. The four end-member water column profile types used to create the structured principal component axes are indicated on both plots. All glider profiles collected in 2013 and 2014 are also shown (grey dots). Glider profiles from the SPOT location are shown in black diamonds and compared against ship-based profiles (grey and black squares).

**Figure 4:** Temperature and chlorophyll profiles for end-member profiles. Average temperature and chlorophyll a profiles for the four end-member profile types in the San Pedro Channel during spring and early summer in 2013 and 2014: cool, low chlorophyll (n=12), cool, high chlorophyll (n=10), warm, subsurface high chlorophyll (n=15), and warm, low chlorophyll (n=17). Average mixed layer depth for each end-member profile is denoted with a circle. Secondary characteristics from these four water column profile types were used to create the structured principal component axes for downstream analyses.

**Figure 5:** Cross-channel coherence. SPOT profiles were grouped by PCA quadrant. Panel (a) displays SPOT profiles (diamonds) and the corresponding non-SPOT profiles (grey dots) that were collected during the same glider transect when SPOT was most similar to the CLC end-member. Similarly, panels (b), (c), and (d) display the profiles for which SPOT was most similar to the WLC, CHC, and WSHC end-members, respectively.

**Figure 6:** Seasonal progression of bins along the transect. Two examples are shown: SPOT (bin 28) and nearshore bins (average for bins 55-62). The average and standard deviation for April (blues) and June (greens) are shown. Both sites show a transition from low PC1 values to elevated PC1 values indicating increased offshore influence.

**Figure 7:** Integrated primary production (PP) within the first optical depth (OD1) versus integrated primary production over the euphotic zone. Each profile is colored by its PC2 value. The trendlines for all profiles falling within the four end-member profile clusters are shown.

**Figure 8:** Impact of sampling frequency. Panel a shows the impact of different sampling frequencies on the estimated SPOT water column profile characteristics. The mean and standard deviation of full glider dataset for SPOT (~4 day sampling, 1 dataset) from March-July 2013 and 2014 are shown in black. The mean and standard deviation of the subsampled glider dataset for the SPOT site (8-day, 2 datasets; 12-day, 3 datasets; 16-day, 4 datasets; 20-day, 5 datasets; 24-day, 5 datasets) are shown as colored circles. For reference, the mean and standard deviation of the ship-based SPOT time-series samples from

March -July (Ship, 30), and the ship-based UpRISEE cruise samples from March-July (Ship, 14) are shown. The 95% confidence intervals for the four end-member profiles are also shown. Panel b displays the differences in estimated characteristics for the subsampled datasets relative to the full glider dataset (4-day sampling).


**Figure 9:** Interannual variability in SPOT cruise profiles versus high-resolution glider profiles. The mean and standard deviation of the PC1 and PC2 values for the SPOT cruise profiles from 2000 to 2011 are shown. The mean (dashed line) and standard deviation (grey shading) of the 2013 and 2014 glider profiles are also indicated.

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

Table 1: Distribution of profile types as estimated by the structured PCA. The number of profiles that fall within with each end-member group (95% confidence interval) is given (N, %) for: all glider profiles (all glider), glider profiles from the SPOT bin (SPOT glider), all ship-based SPOT profiles (SPOT cruise), ship-based SPOT profiles from the months that the gliders were in the water (SPOT cruise March-July), all UpRISEE ship-based profiles (UpRISEE cruise), and UpRISEE profiles from the months that the gliders were in the water (UpRISEE cruise March-July).

| | All Glider Samples (March-July) | | SPOT Glider Samples (March-July) | | SPOT Cruise Samples (All Months) | | SPOT Cruise Samples (March-July) | | UpRISEE Cruise Samples (All Months) | | UpRISEE Cruise Samples (March-July) | |
|---|---|---|---|---|---|---|---|---|---|---|---|---|
| | N | % | N | % | N | % | N | % | N | % | N | % |
| Total Sample Size | 1606 | -- | 54 | -- | 64 | -- | 30 | -- | 21 | -- | 14 | -- |
| Cold High-Chlorophyll | 19 | 1.2% | 1 | 1.8% | 0 | 0.0% | 0 | 0.0% | 0 | 0.0% | 0 | 0.0% |
| Warm Sub-surface High Chlorophyll | 398 | 24.8% | 14 | 25.9% | 10 | 15.6% | 2 | 6.6% | 4 | 19.0% | 3 | 21.4% |
| Warm Low-Chlorophyll | 194 | 12.1% | 5 | 9.3% | 8 | 12.5% | 4 | 13.3% | 4 | 19.0% | 3 | 21.4% |
| Cold Low-Chlorophyll | 25 | 1.6% | 0 | 0.0% | 0 | 0.0% | 0 | 0.0% | 0 | 0.0% | 0 | 0.0% |
| Not Within 95% C.I. | 970 | 60.4% | 34 | 63.0% | 46 | 71.8% | 24 | 80.0% | 13 | 61.9% | 8 | 57.1% |

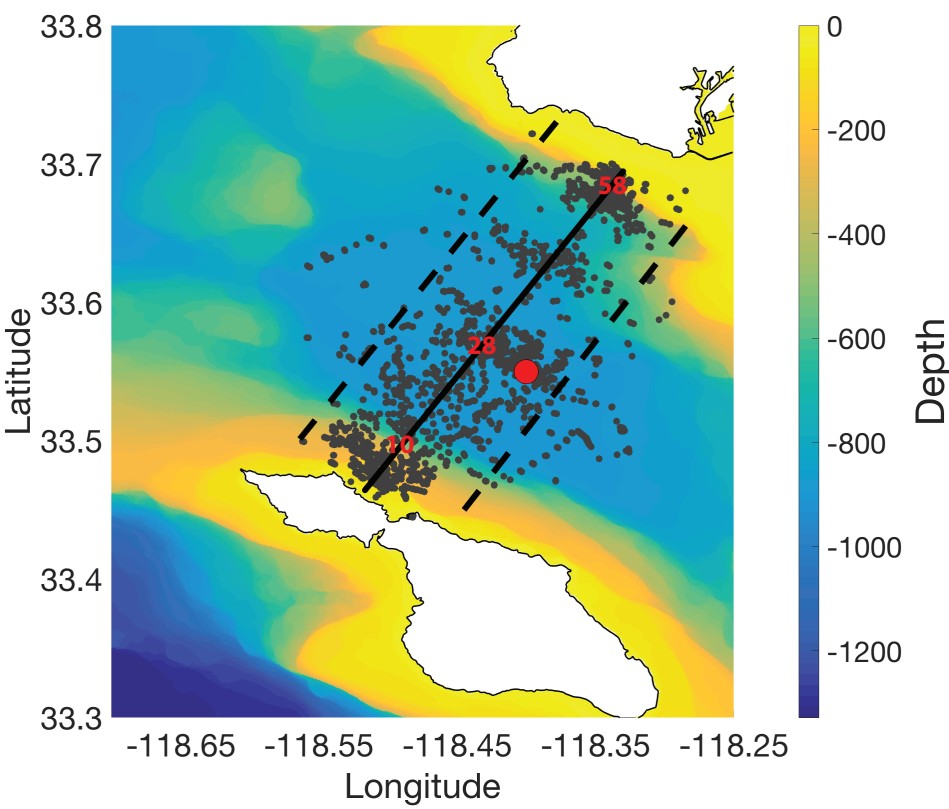

**Figure 1:** Glider deployment map and idealized glider transect. Slocum electric gliders were deployed in 2013 and 2014 between the Palos Verdes Peninsula and Catalina Island in the San Pedro Channel. The San Pedro Ocean Time-series (SPOT) station, located at 33° 33' 00" N and 118° 24' 00" W, is indicated by the red dot. Glider surfacings are indicated with grey dots. An idealized transect was defined running perpendicular to the mean flow (thick black line). Glider profiles collected within 5 kilometers of the idealized transect line (dashed black lines) were used to assess cross-channel variability of oceanographic properties during these glider deployments. The location of bin 10, 28 (closest to SPOT), and 58 are shown in red. The distance from bin 10 to Catalina Island is 5.3 km, from bin 10 to bin 28 is 9 km, from bin 28 to 58 is 15 km, and from bin 58 to the closest point on the mainland is 4.8 km. The bathymetry of the study area is designated by the color contours and ranged from ~ 20 - 900 m. The maximum glider dive depth was 90 m.

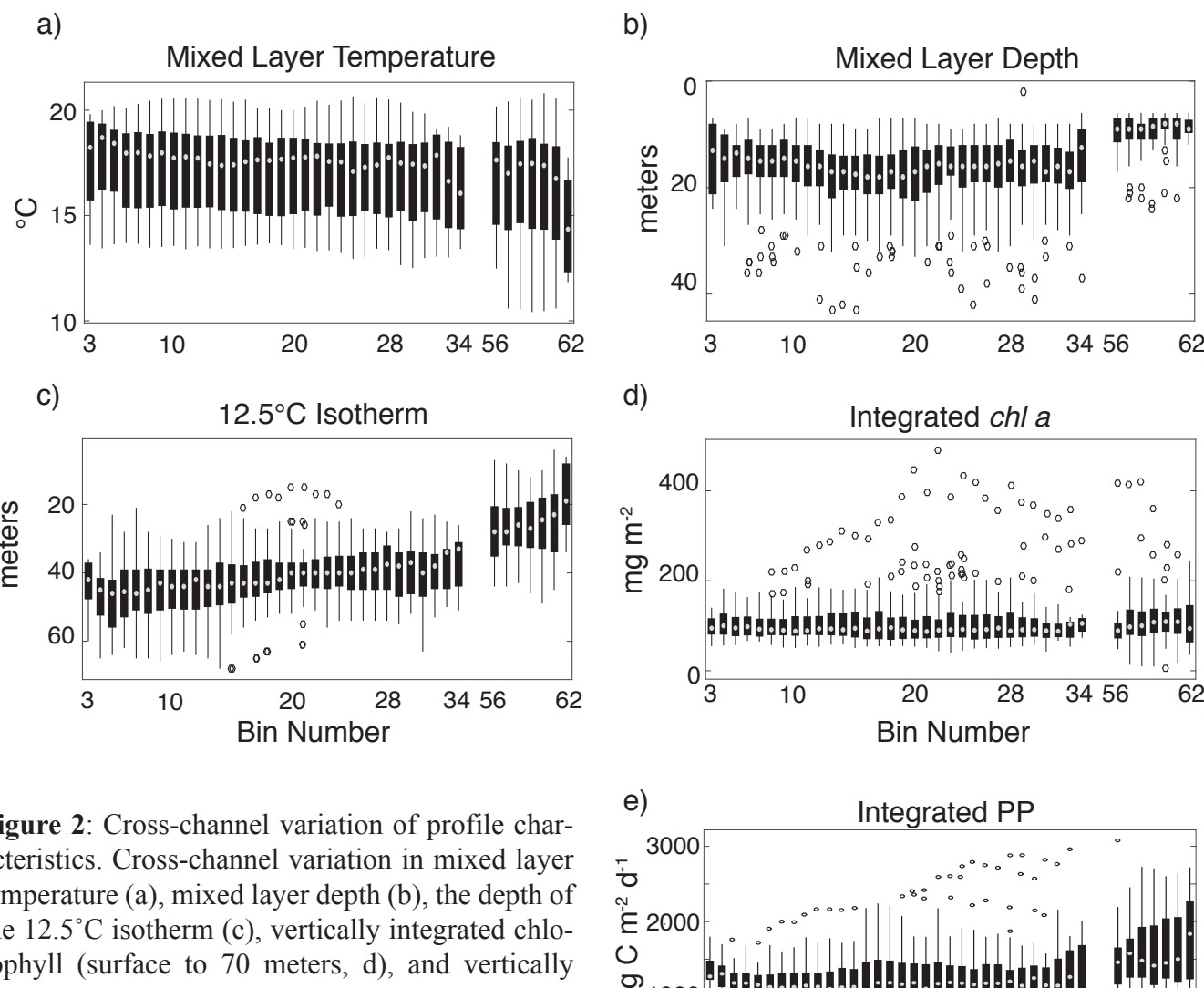

**Figure 2**: Cross-channel variation of profile characteristics. Cross-channel variation in mixed layer temperature (a), mixed layer depth (b), the depth of the 12.5°C isotherm (c), vertically integrated chlorophyll (surface to 70 meters, d), and vertically integrated primary production (surface to euphotic depth, e) for gridded glider profiles from March through July of 2013 and 2014 are shown. Cross-channel box plots show the median value for each bin (white dot), data between the 25th and 75th percentiles (black box), data between the 9th and 91st percentiles (black lines), and outliers (black circles). Low numbered bins correspond with the western side of the San Pedro Channel (SPC), near Catalina. High numbered bins correspond with the eastern side of the SPC, near the Palos Verdes Peninsula (PV). Bins 35-55 correspond with the shipping lanes for the Port of Los Angeles and so have been removed due to incomplete profiles (<85%).

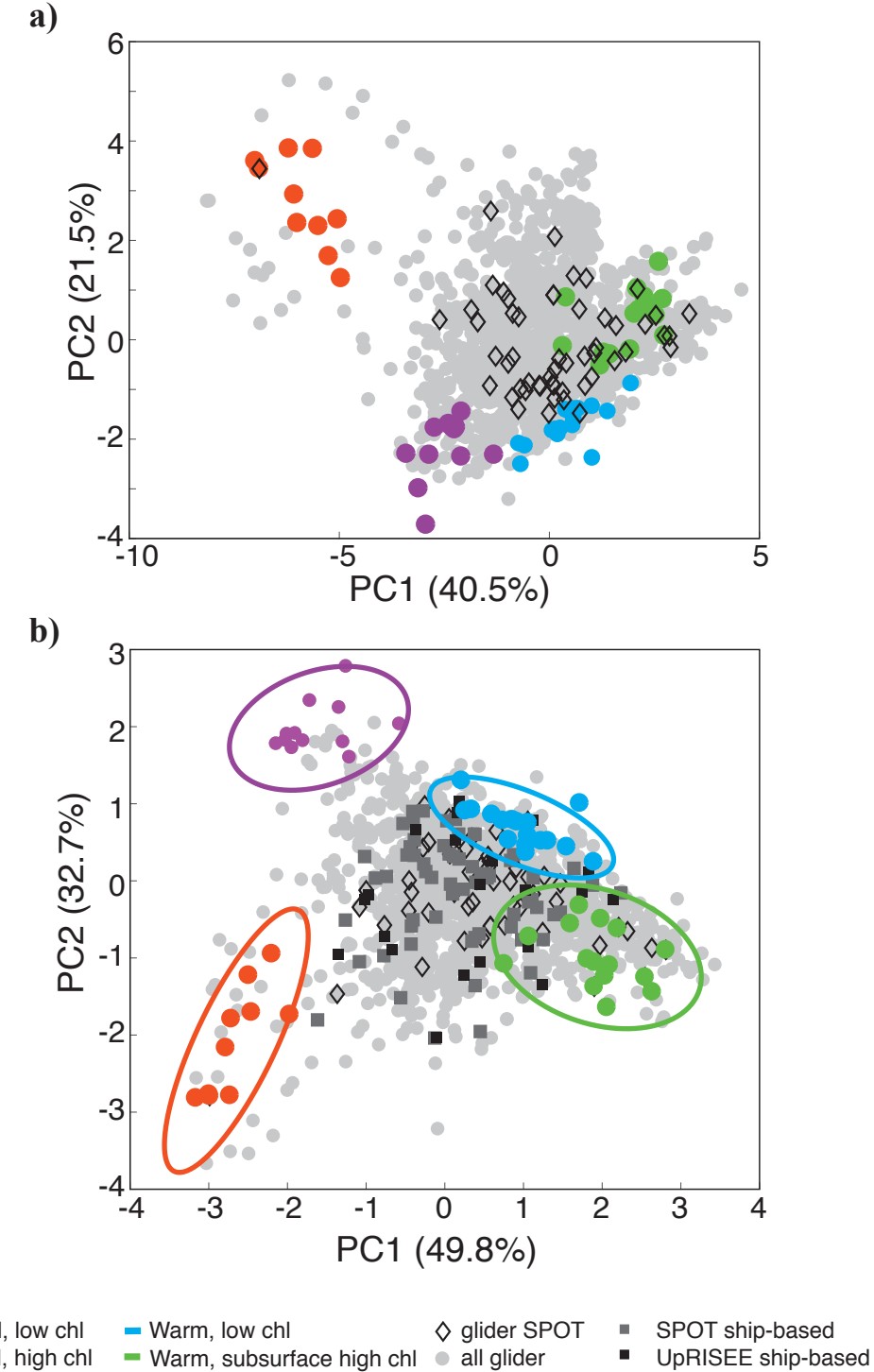

**Figure 3:** Principal Component Analysis of glider profiles. The original PCA (a) and structured PCA (b) are shown. The four end-member water column profile types used to create the structured principal component axes are indicated on both plots. All glider profiles collected in 2013 and 2014 are also shown (grey dots). Glider profiles from the SPOT location are shown in black diamonds and compared against ship-based profiles (grey and black squares).

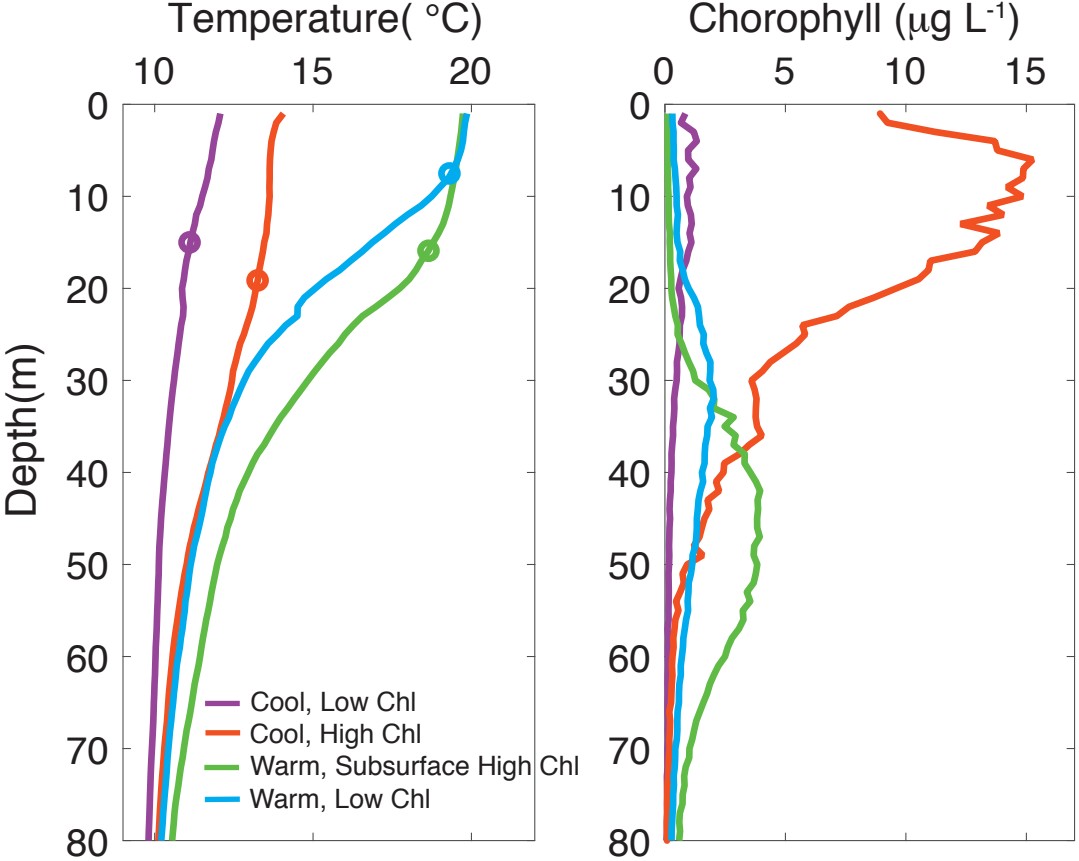

**Figure 4:** Temperature and chlorophyll profiles for end-member profiles. Average temperature and chlorophyll a profiles for the four end-member profile types in the San Pedro Channel during spring and early summer in 2013 and 2014: cool, low chlorophyll (n=12), cool, high chlorophyll (n=10), warm, subsurface high chlorophyll (n=15), and warm, low chlorophyll (n=17). Average mixed layer depth for each end-member profile is denoted with a circle. Secondary characteristics from these four water column profile types were used to create the structured principal component axes for downstream analyses.

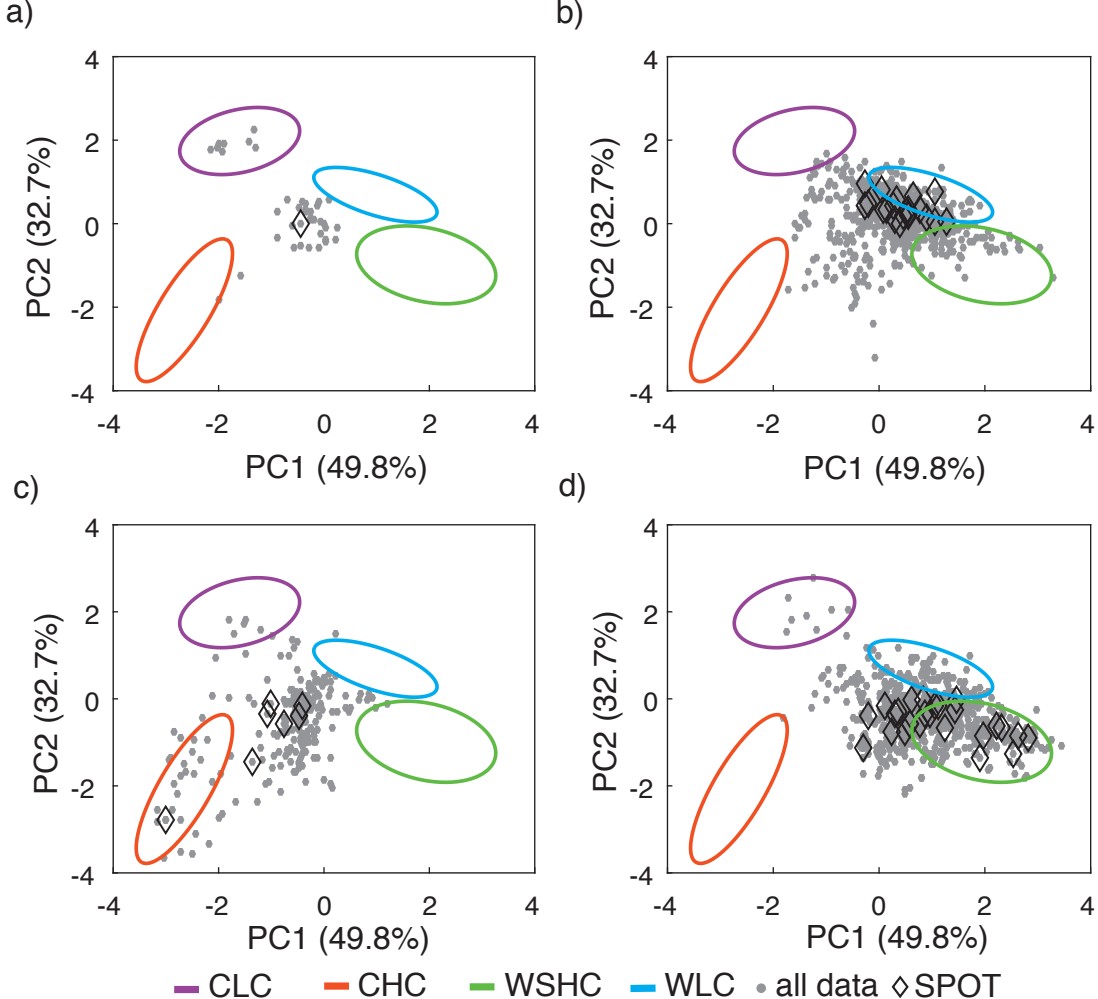

**Figure 5:** Cross-channel coherence. SPOT profiles were grouped based on PCA quadrant. Panel (a) displays SPOT profiles (diamonds) and the corresponding non-SPOT profiles (grey dots) that were collected during the same glider transect when SPOT was most similar to the CLC end-member. Similarly, panels (b), (c), and (d) display the profiles for which SPOT was most similar to the WLC, CHC, and WSHC end-members, respectively.

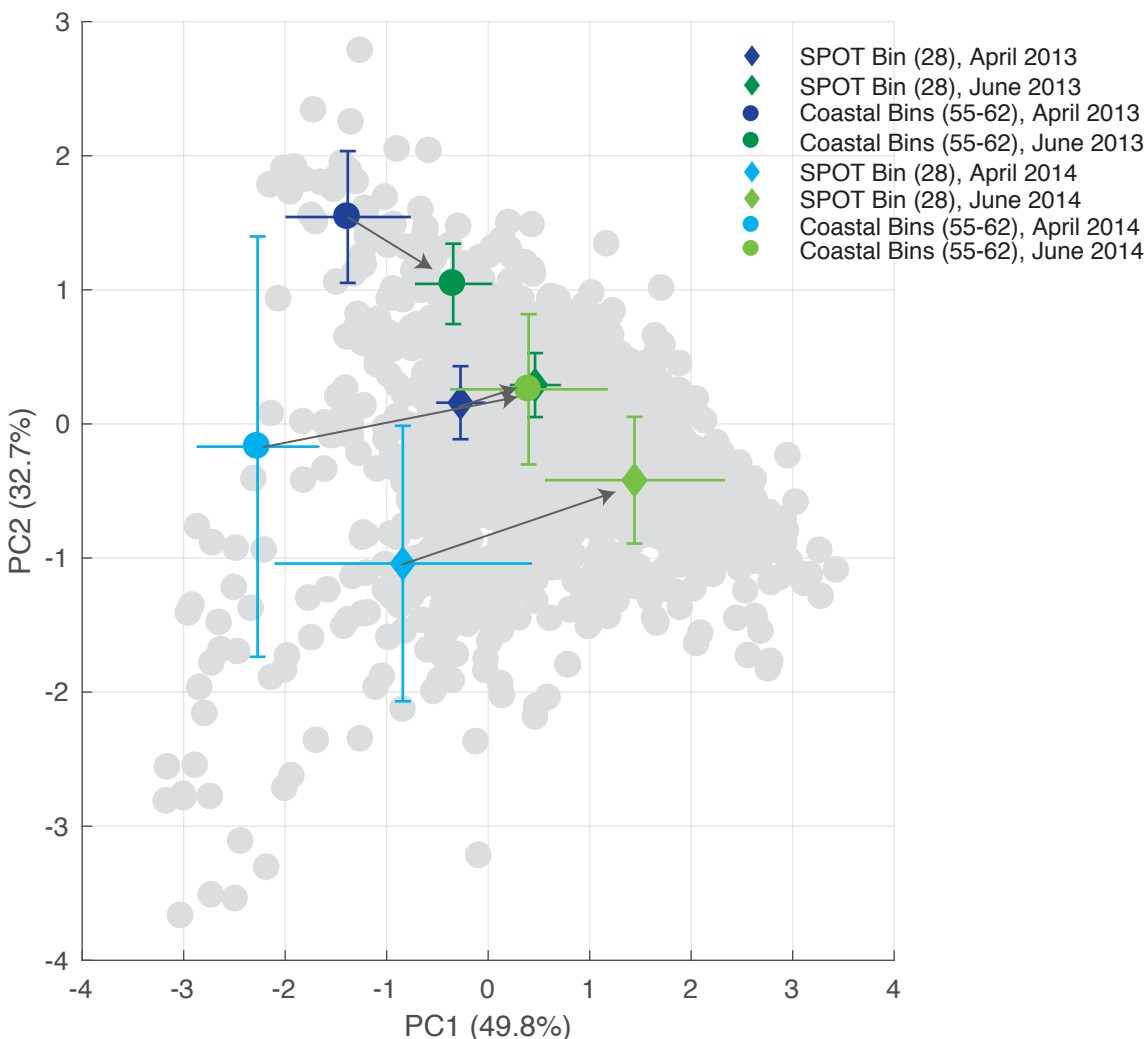

**Figure 6:** Seasonal progression of bins along the transect. Two examples are shown: SPOT (bin 28) and nearshore bins (average for bins 55-62). The average and standard deviation for April (blues) and June (greens) are shown. Both sites show a transition from low PC1 values to elevated PC1 values indicating increased offshore influence.

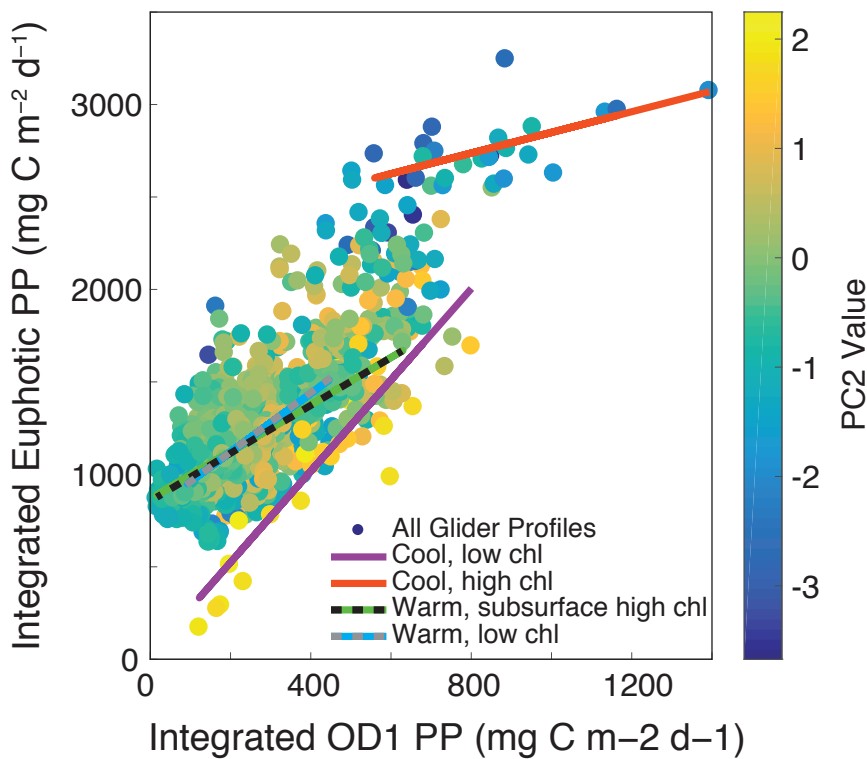

**Figure 7:** Integrated primary production (PP) within the first optical depth (OD1) versus integrated primary production over the euphotic zone. Each profile is colored by its PC2 value. The trendlines for all profiles falling within the four end-member profile clusters are shown.

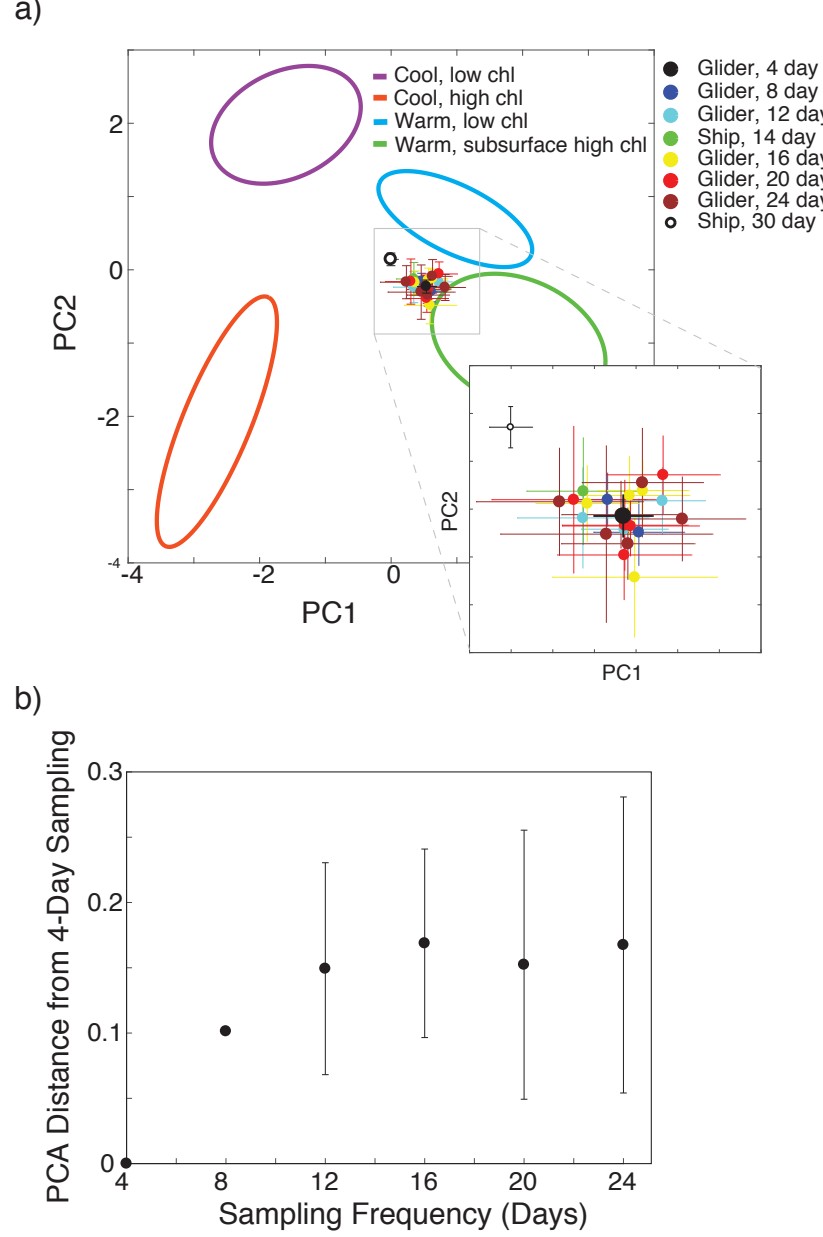

**Figure 8**: Impact of sampling frequency. Panel a shows the impact of different sampling frequencies on the estimated SPOT water column profile characteristics. The mean and standard deviation of full glider dataset for SPOT(~4 day sampling, 1 dataset) from March-July 2013 and 2014 are shown in black. The mean and standard deviation of the subsampled glider dataset for the SPOT site (8-day, 2 datasets; 12-day, 3 datasets; 16-day, 4 datasets; 20-day, 5 datasets; 24-day, 5 datasets) are shown as colored circles. For reference, the mean and standard deviation of the ship-based SPOT time-series samples from March -July (Ship, 30), and the ship-based UpRISEE cruise samples from March-July (Ship, 14) are shown. The 95% confidence intervals for the four end-member profiles are also shown. Panel b displays the differences in estimated characteristics for the subsampled datasets relative to the full glider dataset (4-day sampling).

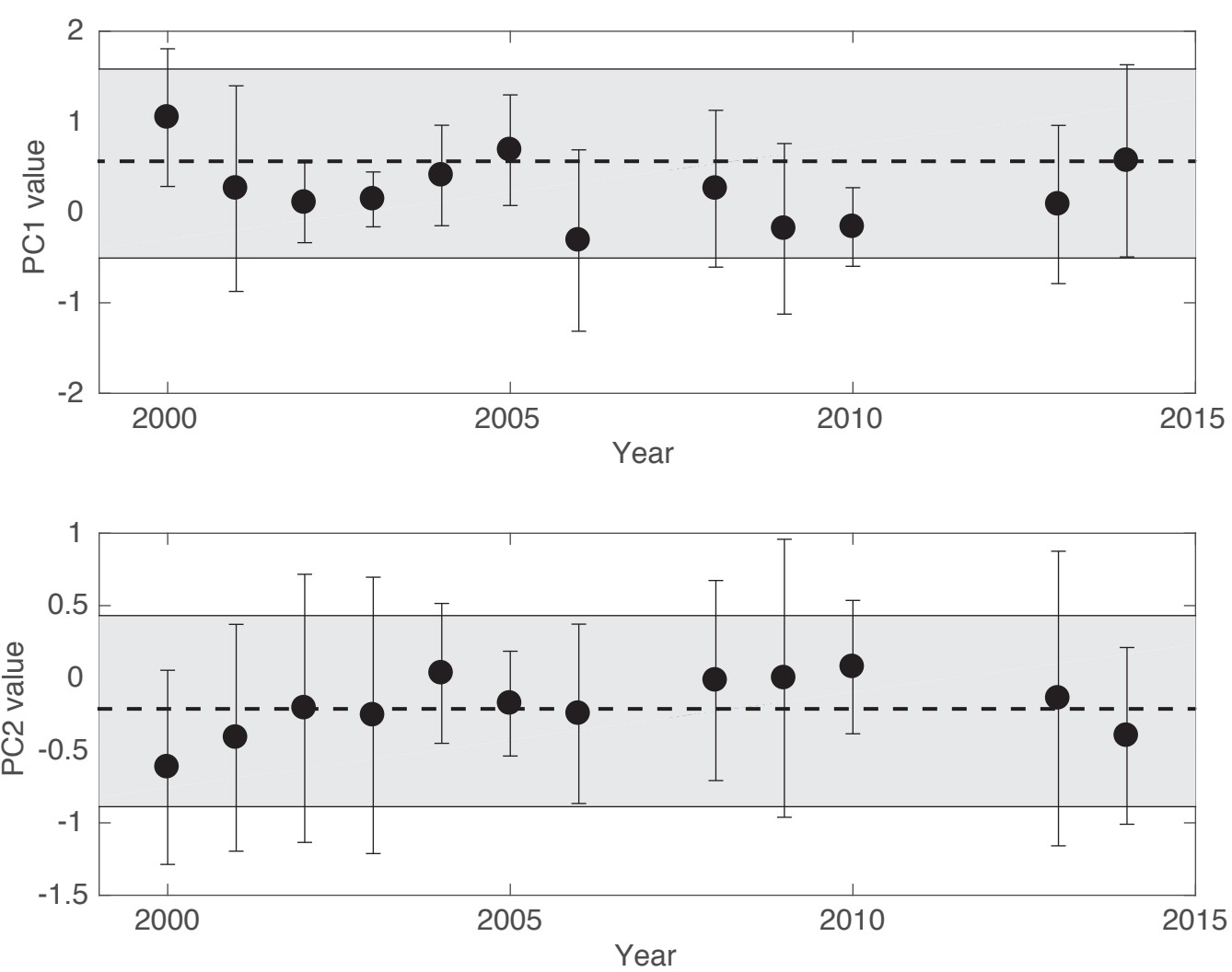

**Figure 9**: Interannual variability in SPOT cruise profiles versus high-resolution glider profiles. The mean and standard deviation of the PC1 and PC2 values for the SPOT cruise profiles from 2000 to 2011 are shown. The mean (dashed line) and standard deviation (grey shading) of the 2013 and 2014 glider profiles are also indicated.