# Peer review of "Contextualizing time-series data: Quantification of short-term regional variability in the San Pedro Channel using high-resolution *in situ* glider data"

_Biogeosciences, 2017_

## Referee Comment (RC1) · Anonymous Referee #1 · 12 Feb 2018

The manuscript by Teel et al. present a dataset of T, S and chlorophyll-a obtained from glider crossings in the San Pedro Channel, in the Southern California Bight. The channel is home to the San Pedro Ocean Time Series (SPOT), a long running oceanographic station that is sampled about once a month. Teel et al. analysis of the glider dataset suggests that SPOT profiles are overall representative of the SPC, with conditions ranging from oligotrophic deep chlorophyll maxima similar to offshore waters (by far the dominant pattern), to post-upwelling surface blooms. Notably, the glider data shows very weak correlations with satellite-based estimates of the surface chlorophyll,

raising doubts about reliability of satellite data in the region. Ocean time series have been fundamental in advancing our understanding of ocean physics and biogeochemistry. However, they are localized in space, and sampled at most at monthly frequencies, raising the question of how representative they may be for broader regions, and how much high-frequency variability they may miss. This is a critical question in a region like the SCB, where complex circulation patterns, including upwelling, mesoscale eddies, island wakes, sustain variability on a range of timescales. Thus the work by Teel et al., is a welcome attempt at characterizing variability at a time series in relation to larger scales. I am of two minds about the paper. The dataset presented is of good quality and potentially useful in elucidating physical and biogeochemical variability in the region. In fact, I encourage the Authors to make the data available for the community. The decomposition of this variability into representative modes is also a useful insight. However, some aspects of the methods, the presentation of the results, and some parts of the discussion are not very clear, and made for a difficult and often opaque read. I encourage the Authors to work on a better synthesis and explanation of their results.

General comments:

Footprint of SPOT data. The paper wants to make a broad claim about the variability at SPOT, but the glider dataset has itself quite a limited footprint, extending for one 28 km between the mainland and Catalina Island. This is scarcely representative of the broader Southern California Bight, and it would have been nicer to have a broader sampling, or comparison with a broader set of observations. This is especially important in light of few of the main aims of the paper, e.g. addressing local vs. regional drivers of variability (abstract, line 19), and determining the spatial domain of a time series (abstract, line 28). I feel that more effort could have been made to discuss how the study resolves these questions, at least for the SPC. After reading the paper, I am not sure I have a clearer idea of the questions.

Actual SPOT data. While the data discusses at depth the representativeness of the

SPOT time series for the context of the SPC, actual profiles from SPOT dataset are not used, but only glider data that pass through the SPOT station. I think there is a missed opportunity to for a reanalysis of the SPOT data in light of the information provided by the new glider dataset. With ∼monthly sampling, nearly 20 years of SPOT data should contain ∼100 profiles (for March-July periods) that could be easily couched within the variability identified by the glider profiles. Do they all fall within the range of variability observed here? Are the frequencies of the different modes observed in the SPOT data in agreement with their frequencies from the glider data analysis? It would be interesting to know if there are outlier in the SPOT data, which may suggest perhaps importance of inter annual variability.

Definition of "end members". The separation of the variability into main modes is a good idea, but I have some criticisms on the way it is conducted and presented. The modes are identified a priori in a somewhat arbitrary way, which is not very well described. I would think an objective (i.e. replicable) approach could have been running a PCA of the entire dataset, then extracting the main axes of variation and use extreme values to define "end-members". Here it seems the Authors qualitatively selected 54 profiles, then identified PCA for them, and projected the entire dataset on the resulting PCA. (In fact, I think the entire methods are not clear, and deserve a dedicated, more detailed section, which could go in the Supplementary Information.) Now, I think expert judgment is often a reasonable approach, but more discussion of the rational between end-members and their translation to the whole dataset should be presented. Also, by looking at Fig. a, the "offshore influence" mode could be considered, rather than an end-member, somewhat a mixture of "early upwelling" and "deep chl max" based on PCA values. Is it really an end-member?

I also have a quibble about the names of the end members: I kept confusing "offshore influence" and "deep chlorophyll maximum". The two names make me both think of offshore oligotrophic subtropical conditions, and it took me a while to realize that "offshore influence" is in fact in between offshore oligotrophic and coastal influences. The

"deep chlorophyll maximum" end-member is in fact more representative of the offshore regions than the "offshore influence". Maybe a better naming strategy can be found.

Comparison between glider and satellite-based estimates of surface chlorophyll-a. This is extremely interesting, and may contain some of the most relevant implications of the study for a broader community. The mismatch between surface glider data and satellite retrieval is glaring (supplementary Fig. S4), and suggest that satellite data should be taken very cautiously in the nearshore SCB, or even completely discarded as a reliable source of information on phytoplankton distribution. The Authors even state in line 278 and in the caption of Fig. S4 that "no correlation was observed between glider and satellite derived integrated chlorophyll": this result seems important enough to require a dedicated figure, at least in the Supplement. At the same time I am not completely convinced of the strength of the Authors' comparison. The Authors do not really get to the bottom of the mismatch, and some of the hypothesis that they put forward don't seem to be able to explain it, especially in light of the systematic variation shown in Fig. S4. I suspect some systematic mismatch in the optical depth over which the glider data should be integrated to provide comparison with the satellite data may be behind the discrepancies. Also, are satellite algorithms really only representative of the first optical depth? Given the exponential nature of light-attenuation in water, perhaps satellite retrievals of ocean color may be representative of a somewhat deeper water column. I think the Authors identify an important issue, but I am not sold it is time to start ignoring satellite retrievals of chlorophyll-a in the region.

The use of the concept of "connectivity" for both horizontal and vertical similarities is somewhat misleading. The fact that inshore and offshore profiles may be similar doesn't necessary imply a direct, material bath connecting the two, as the word connectivity implies, but they may be just responding to remote, synchronous variations that occurs at scales large than few tens of km sampled by the glider. I suggest using a term different than connectivity throughout the text, especially section 3.2 and 3.3. Perhaps "coherence" or "similarity" would be more accurate.

Some of the results could be less vague and speculative, and more quantitative. For example, Line 150 "given sufficient sampling, SPOT data could be representative of the average state of the SPC": can "sufficient sampling" be actually quantified? Similarly, line 355 in the conclusions, "higher frequency sampling": can this be quantified based on the new data? Would weekly sampling be needed? or daily? Specific comments:

Line 70, "recurring membership": please clarify or rephrase the term.

Line 92, "that was perpendicular to the mean flow": add "approximately"

Line 105, calculation of BVF. This requires a vertical derivative, which dramatically increase the noise in the resulting variable. Was any smoothing applied to the data to reduce the noise?

Line 112, conflation of depth of 12.5C isotherm and nutricline. I wonder if this relationship could be tested with SPOT data for the region. Are nutrients measured at SPOT? How well does the relationship hold there?

Line 126: the "and" after "data" seems out of place.

Line 141-142, "This tilt is consistent with equatorward flow through the channel". This statement is at odds with a generally poleward flow in the very nearshore band and in the SPC, which tends to bring waters from the southern bight, and is embedded in the Southern California Eddy. The predominantly equatorward California Current is much more offshore.

Line 170, "seasonal traits" clarify o rephrase.

Line 208-209, "we cannot distinguish between local and remote sources": this is at odds with the previous sentence, and with the general notion that the glider data allows a characterization of the spatial domain of the time series data. Please clarify.

Line 214, "maintained an onshore-offshore gradient on average": please clarify the sentence.
[Figure]

Line 244, "highly offshore characteristics (Figure 6a)": this seems to refer to the wrong panel, please double check.
 Line 249-250: this sentence seems to undermine the idea of connectivity as a material path connecting inshore and offshore.

Line 288, "bloom thickness": defined how?

Line 321, "decreased sedimentation": please clarify.

Line 331, "events": the term doesn't seem appropriate for the modes considered, especially deep chlorophyll maxima which seem the common "background" state for the SPC. Please clarify.

Figure 1. Some bin numbers could be useful, e.g. 10,20,58, since they are used later.

Figure 2. The mean mixed layer could be shown on this figure too for the 4 end-members.

Figure 3. It would be useful to have a measure of the distance from the Palos Verdes Peninsula together with the bin numbers in the x-axis. Also, it would be useful to add the location of SPOT in the panels. The caption should also specify the months of the observations, besides the years.

Figure 4. Are two panels really necessary? It seems that all information of panel b could be contained in panel a. Also, how were the ellipses defined?

Figure 5 is a bit messy, and could go into the Supplement. It is not very successful in showing a clear seasonal progression or cycle, like the similar figures in Jacox, 2016 that presumably inspired it. To me the message is that the potential seasonal progression of upwelling is swamped by high frequency variability. (Or that perhaps the 2 PCA axes are not well positioned to highlight this progression.)

Figure 7: I wonder if the ratio between surface and total integrated cha may be a better variable to show here.

Supplementary Table S1: many of the thresholds and combinations behind the endmember definition seem somewhat arbitrary, and could be better justified, e.g. with a dedicated Supplement section.

Supplementary Figure S2: what are the vectors on the plot? Please explain in the caption.

Supplementary Figure S3: please correct the units in the caption of panel a, from ug to mg/m3.

Supplementary Figure S4: the legend of the figure states "Satellite:Glider" mach-ups, but the labels and caption state "Glider-Satellite", please clarify.

[Figure]

---

## Referee Comment (RC2) · Anonymous Referee #2 · 3 Mar 2018

This paper uses high-frequency spatial and temporal glider data to quantify variability at the coastal San Pedro Ocean Time-series (SPOT) site in the San Pedro Channel (SPC) and provide insight into the underlying oceanographic dynamics for the site.

The glider data (a total of 1606 profiles) collected from March through July of 2013 and 2014 are used. This is a very rich data set and a detailed analysis is well justified for a publication. However, the manuscript in its current form is very difficult to read and follow. PCA is used to differentiate different profile types. It is confusing how the 54 end-member profiles are selected to define each of four dominant profile types, and

then the remaining 1552 profiles are then projected onto the PC1 and PC2 coordinates. Maybe a more detailed description of the methodology is needed in the supplemental information.

Time series are mentioned as the motivation of this paper, although the SPOT data are not used in the analysis. Both weekly and monthly time scales are mentioned in the text, what is the time interval for the SPOT measurements? It is not true "most time-series are sampled ... approximately once per month. Many time series use mooring platforms collecting data every few minutes.

p2, end of the 1st paragraph, "...at an individual site relative to a larger region may provide a path for leveraging numerous local time series sites in order to gain an understanding of larger scale oceanographic dynamics." What is the spatial scale for this "larger" region/scale? Maybe the SPOT time series can be used to quantify this spatial scale.

p2, 2nd paragraph, "cloud contamination" is not mentioned as the primary reason to have limited coverage.

"coastal and offshore processes", define "coastal" and "offshore"

It seems arbitrary to have the four dominant water column profile types: early upwelling, surface phytoplankton bloom, subsurface chlorophyll maximum, and offshore influence. Again define "offshore" here. Should the wind forcing be used?

p5, satellite data are mentioned, but should be used more to study the surface and subsurface linkage

p12, 5. Conclusion, end of the 1st paragraph, "...insensitive to coastal anthropogenic change...well positioned to identify a regional response to climate change." how do you derive such a conclusion?

Table 1, define "SPOT specific profiles", "SPOT samples", what is CI?

Figure 1, I understand the color represents bathymetry, why don't you state this in the caption?

Figure 2, what is the arrows mean below the figure, PC1, PC2? what does the "n=" mean?

Figure 3, is "box plot" a more standard term than "whisker plot", see https://en.wikipedia.org/wiki/Box_plot; define "bin"

Supplemental Figure S2, define "ideal profiles"

---

## Author Comment (AC1) · 3 Apr 2018

*The manuscript by Teel et al. present a dataset of T, S and chlorophyll-a obtained from glider crossings in the San Pedro Channel, in the Southern California Bight. The channel is home to the San Pedro Ocean Time Series (SPOT), a long running oceanographic station that is sampled about once a month. Teel et al. analysis of the glider dataset suggests that SPOT profiles are overall representative of the SPC, with conditions ranging from oligotrophic deep chlorophyll maxima similar to offshore waters (by far the dominant pattern), to post-upwelling surface blooms. Notably, the glider data shows very weak correlations with satellite-based estimates of the surface chlorophyll, raising doubts about reliability of satellite data in the region. Ocean time series have been fundamental in advancing our understanding of ocean physics and biogeochemistry. However, they are localized in space, and sampled at most at monthly frequencies, raising the question of how representative they may be for broader regions, and how much high-frequency variability they may miss. This is a critical question in a region like the SCB, where complex circulation patterns, including upwelling, mesoscale eddies, island wakes, sustain variability on a range of timescales. Thus the work by Teel et al., is a welcome attempt at characterizing variability at a time series in relation to larger scales. I am of two minds about the paper. The dataset presented is of good quality and potentially useful in elucidating physical and biogeochemical variability in the region. In fact, I encourage the Authors to make the data available for the community. The decomposition of this variability into representative modes is also a useful insight. However, some aspects of the methods, the presentation of the results, and some parts of the discussion are not very clear, and made for a difficult and often opaque read. I encourage the Authors to work on a better synthesis and explanation of their results.*

To address the overall concerns raised by the Reviewer, we have edited the text to clarify the methods, added additional analyses, and included a comparison between the glider data and ship-based SPOT measurements. Below we provide a detailed point-by-point response to each specific comment. We are fully committed to making this data available to the community and plan to upload the glider data used in this analysis to the BCO-DMO repository (https://www.bco-dmo.org/) upon publication of this work.

*General comments:*
*Footprint of SPOT data. The paper wants to make a broad claim about the variability at SPOT, but the glider dataset has itself quite a limited footprint, extending for one 28 km between the mainland and Catalina Island. This is scarcely representative of the broader Southern California Bight, and it would have been nicer to have a broader sampling, or comparison with a broader set of observations.*

We completely agree that additional data with a wider footprint would be fantastic and would help constrain variability within the Southern California Bight. Unfortunately, we were limited in the number of gliders we had access to and the time in which they could be in the water. That said, the cross-channel glider transects were strategically chosen in order to best sample the

region. As flow is primarily north-south through the San Pedro Channel, the largest gradients are expected east-west and so the gliders were focused on capturing these features (Hickey, 1992; Noble et al., 2002; Di Lorenzo, 2003; Hickey et al., 2003; Noble et al., 2009b). In addition, the deployment times were chosen to span the seasonal upwelling events, again with the aim to best capture variability within the San Pedro Channel (Hickey, 1992; Di Lorenzo, 2003; Hickey et al., 2003). We have edited the text to clarify the choice of the transect path and the deployment time and how these choices might impact estimates of variability in the San Pedro Channel. We have also added a discussion on the channel as a proxy for the larger Southern California Bight.

The primary aim of this work was to leverage high temporal and spatial scale data (e.g. glider data) to generate a framework for understanding datasets sampled with coarser temporal and spatial resolution (e.g. monthly time-series). We hope that this conceptual framework will be applied to other datasets for the region in order to compare variability across the larger California Current System.

*This is especially important in light of few of the main aims of the paper, e.g. addressing local vs. regional drivers of variability (abstract, line 19), and determining the spatial domain of a time series (abstract, line 28). I feel that more effort could have been made to discuss how the study resolves these questions, at least for the SPC. After reading the paper, I am not sure I have a clearer idea of the questions.*

We have edited the text to help clarify the primary conclusions of the paper. Specifically, our analysis of the glider-based high-resolution data shows that the SPOT time-series data are more reflective of the offshore stratified environment rather than the near-coastal upwelling environment where coastal discharge (outfalls and storm water) are a more significant factor. As such, we conclude that the SPOT site is more likely to be influenced by regional responses to climatic shifts than to local events (e.g. increased discharge into the port of Los Angeles). In addition, we show that the end-member PCA provides a useful framework for analyzing seasonal and interannual shifts in variability at a time-series site.

*Actual SPOT data. While the data discusses at depth the representativeness of the SPOT time series for the context of the SPC, actual profiles from SPOT dataset are not used, but only glider data that pass through the SPOT station. I think there is a missed opportunity to for a reanalysis of the SPOT data in light of the information provided by the new glider dataset. With ~monthly sampling, nearly 20 years of SPOT data should contain ~100 profiles (for March-July periods) that could be easily couched within the variability identified by the glider profiles. Do they all fall within the range of variability observed here? Are the frequencies of the different modes observed in the SPOT data in agreement with their frequencies from the glider data analysis? It would be interesting to know if there are outlier in the SPOT data, which may suggest perhaps importance of inter annual variability.*

We thank the Reviewer for this suggestion. We have added a comparison between both ship-based SPOT data from 2000-2011 and data from a set of Upwelling Regime In-Situ

Ecosystem Efficiency study (UpRISEE) cruises that occurred during the time of the glider deployments (2013-2014) (*see revised Figure 4*). This allows us to both analyze interannual variability and seasonal differences in variability. Specifically, we find that the variability in water column profiles captured by the high-resolution gliders during the upwelling period (March-July) of 2013-2014 was comparable to the interannual and seasonal variability observed in monthly sampling at SPOT over a decade (*Figure R1*). In addition, samples taken every two-weeks during the UpRISEE period also matched the glider profiles from SPOT (*Figure R2*). We have modified Figure 4 and updated Table 1 to include these ship-based measurements (*see attached*).

[Figure]

Figure 4: Structured Principal Component Analysis of glider and ship-based profiles. The four end-member water column profile types (Supplemental Figure S2) were used to create principal component axes. Physical variability was associated with PC1 (49.8% of total variance) and biological variability was associated with PC2 (32.7% of total variance). Panel A shows all data projected onto these axes including: all glider profiles collected in 2013 and 2014 (grey dots), glider profiles from the SPOT location (black diamonds), SPOT ship-based profiles (grey squares), and UpRISEE ship-based profiles (black squares).

[Figure]

Figure R1: Interannual variability in SPOT cruise profiles versus high-resolution glider profiles. The mean and standard deviation of the PC1 and PC2 values for the SPOT cruise profiles from 2000 to 2011 are shown. Also indicated is the mean (dashed line) and standard deviation (grey shading) of the 2013 and 2014 glider profiles. This analysis demonstrates significant interannual variability at the SPOT site but also that this variability falls within the range captured by the glider profiles.

[Figure]

Figure R2: Impact of sampling frequency on the mean and standard error of SPOT profiles in PCA space.

*Definition of "end members". The separation of the variability into main modes is a good idea, but I have some criticisms on the way it is conducted and presented. The modes are identified a priori in a somewhat arbitrary way, which is not very well described. I would think an objective (i.e. replicable) approach could have been running a PCA of the entire dataset, then extracting the main axes of variation and use extreme values to define "end-members". Here it seems the Authors qualitatively selected 54 profiles, then identified PCA for them, and projected the entire dataset on the resulting PCA. (In fact, I think the entire methods are not clear, and deserve a dedicated, more detailed section, which could go in the Supplementary Information.) Now, I think expert judgment is often a reasonable approach, but more discussion of the rational between end-members and their translation to the whole dataset should be presented.*

We have edited the text to clarify how the main modes of variability (end-members) were identified and how the PCA analysis was conducted (*see Extended Methods attached at the end of this response*). In fact, our method is almost exactly what the Reviewer suggests and we hope with the clarified text and expanded methods this is clearer. Briefly, a PCA was conducted without any a priori information and was used to identify the main modes of variability (*Figure R3 attached below*). We than re-ran the PCA using the end-member profiles, defined using a combination of a priori information about the system and the patterns from the original PCA. This allowed us to generate a framework which facilitated meaningful oceanographic interpretation of variability within the glider dataset.

To further clarify the analysis, we have also changed how we refer to the end-member profiles to better clarify what they represent:
*(1) warm, subsurface high chlorophyll (WSHC)*
*(2) warm, low chlorophyll (WLC)*
*(3) cool, low chlorophyll (CLC)*
*(4) cool, high chlorophyll (CHC)*
WSHC represents oligotrophic conditions with an enhanced subsurface chlorophyll maximum, WLC represents oligotrophic conditions with very low biomass throughout the water column, CLC represents early upwelling with cold waters and low chlorophyll, and CHC represents a surface bloom with cool surface temperatures (nominally a coastal bloom).

*Also, by looking at Fig. a, the "offshore influence" mode could be considered, rather than an end-member, somewhat a mixture of "early upwelling" and "deep chl max" based on PCA values. Is it really an end-member?*

We hope that our *Expanded Methods,* including the justification of our choice in end-members, helps address the Reviewer's concern. The 'offshore influence' (now called warm, low chlorophyll) in a purely statistical sense is a mixing between 'early upwelling' (now called cold, low chlorophyll) and 'deep chl max' (now called warm, subsurface high chlorophyll). This also shows up in the default PCA analysis (*Figure R3*). However, based on our understanding of the oceanography of the region, we believe that this profile type is not a mixing of these water masses but is indeed a unique end-member. Specifically, there are two primary physical dynamics that

impact water column signatures in the SPC: 1) coastal upwelling and 2) the Southern California Eddy. Coastal upwelling brings cold high nutrient waters to the surface and triggers large surface blooms that extend from the coast into the channel. The cool, low chlorophyll (CLC) and cool, high chlorophyll (CHC) end-members represent the beginning and end of this process. The Southern California Eddy brings in warm low nutrient waters from offshore. These waters have a distinct signature which we identify here as warm, low chlorophyll (WLC) waters. In fact, of the four end-members, warm, subsurface high chlorophyll (WSHC) is the end-member with the least oceanographic support for being a true, unique end-member. The mechanism that creates periodic elevated subsurface chlorophyll concentrations in this region is still unclear, however there are two leading hypotheses: 1) these are coastal surface blooms that have been advected along isopycnal surfaces out into the channel (e.g., Mitarai et al., 2009; Bialonski et al., 2016; Stukel et al., 2018) or 2) that internal waves result in isopycnal heave of nutrients into the euphotic zone creating enhanced chlorophyll concentrations (e.g., Noble et al., 2009a; Noble et al., 2009b; Lucas et al., 2011). We have added this discussion to the text.

[Figure]

Figure R3: Un-structured Principal Component Analysis of glider profiles. The four end-member water column profile types and the SPOT glider profiles are highlighted. The black arrows denote the the loadings of each variable used in the PCA.

*I also have a quibble about the names of the end members: I kept confusing "offshore influence" and "deep chlorophyll maximum". The two names make me both think of offshore oligotrophic subtropical conditions, and it took me a while to realize that "off- shore influence" is in fact in between offshore oligotrophic and coastal influences. The "deep chlorophyll maximum" end-member is in fact more representative of the offshore regions than the "offshore influence". Maybe a better naming strategy can be found.*

We have altered the names of the end-members to be more descriptive and avoid confusion (*described above*).

*Comparison between glider and satellite-based estimates of surface chlorophyll-a. This is extremely interesting, and may contain some of the most relevant implications of the study for a broader community. The mismatch between surface glider data and satellite retrieval is glaring (supplementary Fig. S4), and suggest that satellite data should be taken very cautiously in the nearshore SCB, or even completely discarded as a reliable source of information on phytoplankton distribution. The Authors even state in line 278 and in the caption of Fig. S4 that "no correlation was observed between glider and satellite derived integrated chlorophyll": this result seems important enough to require a dedicated figure, at least in the Supplement. At the same time I am not completely convinced of the strength of the Authors' comparison. The Authors do not really get to the bottom of the mismatch, and some of the hypothesis that they put forward don't seem to be able to explain it, especially in light of the systematic variation shown in Fig. S4.*

A large body of literature has commented on the relationship between satellite observable chlorophyll (within the first optical depth) and total integrated water column chlorophyll, as well as the need for increased *in situ* sampling to improve satellite chlorophyll and primary production algorithms due to mismatches and inconsistencies in modeling of the *in situ* chlorophyll profiles (e.g., Morel and Berthon, 1989; Stramska and Stramski, 2005; Sathyendranath et al., 1989; Montes-Hugo et al., 2009; Jacox et al., 2015; Seegers et al., 2015). Locally, while satellite surface chlorophyll estimates have been shown to aligned closely with *in situ* glider observations of nearshore surface blooms in the San Pedro Channel, subsurface chlorophyll layers farther offshore were undetected by satellite retrievals (Seegers et al., 2015). Here, we used our framework to identify which oceanographic states for the SPOT site may be most susceptible to satellite misinterpretation. We specifically avoided interpretation of the mismatch between the glider and satellite chlorophyll estimates for a number of reasons. Primarily, we do not find this mismatch surprising given the differences in temporal and spatial scales of these two measurements. Specifically, the glider data were collected at ~0.5km resolution continuously over 24 hours through the deployment while the satellite measurements represent a single pass every 1 to 2 days in the afternoon and averaged over 1 km (Esaias et al., 1998). Furthermore, inaccuracies in the CDOM corrections and atmospheric corrections could also contribute to the observed mismatch (Esaias et al., 1998;Hoge et al., 1999;Wang et al., 2009). We do not feel that our dataset or analysis is the correct one to evaluate the robustness of the MODIS product for the San Pedro Channel and

certainly do not feel that we can conclude that satellite data should be disregarded as a reliable source of information for phytoplankton distributions.

We believe the important take-away from this part of our analysis is that the inherent bias in satellite data of only quantifying chlorophyll over the first optical depth is not a significant issue for samples with high PC2 values and low overall biomass. However, it becomes an increasingly important issue for samples with high biomass and low PC2 values, which have a large percentage of integrated chlorophyll beneath the first optical depth. Specifically, the sub-surface chlorophyll profiles for these samples deviate significantly from the traditional relationship between chlorophyll within OD1 and total integrated chlorophyll. The structured PCA framework provides a metric for assessing the water mass types that may be most problematic for the satellite algorithms. Specifically, our analysis suggests that the satellite vs integrated chlorophyll mismatch may be particularly problematic for some cool, high chlorophyll water mass types (nominally coastal blooms). This suggests that increased *in situ* sampling may be needed when these water mass types are present in order to accurately constrain estimates of biomass distributions and primary production. We have edited the text and Figure 7 to clarify our incorporation of satellite data into our analyses.

[Figure]

Figure 7: Integrated chlorophyll within the first optical depth (OD1) versus total integrated chlorophyll over 70 m for all 2013 and 2014 glider profiles. Each profile is colored by its PC2 value. The location of the four end-member profiles types are also shown.

*I suspect some systematic mismatch in the optical depth over which the glider data should be integrated to provide comparison with the satellite data may be behind the discrepancies.*

We calculated the first optical depth for each glider profile using matched PAR measurement and glider chlorophyll profiles. Specifically, we estimated the light attenuation with depth for each profile as a function of chlorophyll concentration after Jacox et al. (2015). We then calculated the first optical depth as the depth in meters where available PAR is equivalent to $1/e$ of surface PAR after Gordon (1975) and Kirk (1994). The calculations for light attenuation with depth have been validated within the Southern California Bight using *in situ* chlorophyll profiles (Jacox et al., 2015). Using these same light attenuation calculations, glider chlorophyll profiles were used to calculate profile euphotic depths, which were in good correspondence with ship-based euphotic depth measurements from the UpRISEE cruises (Haskell et al., 2016). If anything, our estimate is a conservative one (too deep) because we do not account for additional absorption by any CDOM or for particle backscatter. Therefore, while there are several possible explanations for the satellite-glider mismatched as listed above, we do not believe that there is an issue with our definition of the first optical depth from the glider measurements.

*Also, are satellite algorithms really only representative of the first optical depth? Given the exponential nature of light-attenuation in water, perhaps satellite retrievals of ocean color may be representative of a somewhat deeper water column. I think the Authors identify an important issue, but I am not sold it is time to start ignoring satellite retrievals of chlorophyll-a in the region.*

Based on the literature for remote sensing algorithms, satellite retrievals are an integration over the first optical depth: "The penetration depth of light in the sea is defined for remote sensing purposes as the depth above which 90% of the diffusely reflected irradiance (excluding specular reflectance) originates. It is demonstrated that for a homogeneous ocean, this is the depth at which the downwelling in-water irradiance falls to $1/e$ of its value at the surface." Gordon, (1975). "Satellite-derived retrievals of apparent (AOPs) and inherent (IOPs) optical properties in marine surface waters correspond to a vertically integrated picture of the first optical depth." Montes-Hugo et al., (2009).

*The use of the concept of "connectivity" for both horizontal and vertical similarities is somewhat misleading. The fact that inshore and offshore profiles may be similar doesn't necessary imply a direct, material bath connecting the two, as the word connectivity implies, but they may be just responding to remote, synchronous variations that occurs at scales large than few tens of km sampled by the glider. I suggest using a term different than connectivity throughout the text, especially section 3.2 and 3.3. Perhaps "coherence" or "similarity" would be more accurate.*

We have edited the text to clarify this point. Specifically, we now use the terms 'coherence' and 'covariation' instead of connectivity. We have also added text to discussing the impact of southward advection into the region.

*Some of the results could be less vague and speculative, and more quantitative. For example, Line 150 "given sufficient sampling, SPOT data could be representative of the average state of the SPC": can "sufficient sampling" be actually quantified? Similarly, line 355 in the conclusions, "higher frequency sampling": can this be quantified based on the new data? Would weekly sampling be needed? or daily?*

We have edited the text to be more quantitative where possible. We have also added an analysis of sample frequency using the glider dataset (*see Figure R2 above*). Our preliminary analysis suggests that the majority of the variance at the SPOT site can be captured with sampling every ~14 days, though extreme events were only captured in the highest resolution glider data. However, what is 'sufficient sampling' will be highly dependent on the question being asked. We are working on finalizing this analysis and supporting figure for the revised manuscript.

*Specific comments:*
We have made the requested text edits listed below. Comments are added to answer questions.

*Line 70, "recurring membership": please clarify or rephrase the term.*
*Line 92, "that was perpendicular to the mean flow": add "approximately"*
*Line 105, calculation of BVF. This requires a vertical derivative, which dramatically increase the noise in the resulting variable. Was any smoothing applied to the data to reduce the noise?*
While we believe that the BVF calculation was robust, as we do not use BVF in any of our analyses, we have removed it from the manuscript.

*Line 112, conflation of depth of 12.5C isotherm and nutricline. I wonder if this relationship could be tested with SPOT data for the region. Are nutrients measured at SPOT? How well does the relationship hold there?*
We have confirmed the temperature nutrient relationship at SPOT and confirmed that the 12.5°C isotherm is a good proxy for higher nutrient waters that are within the nitricline (*Figure R4 below*). This relationship also holds within the CalCOFI data over longer time-series as shown by Lucas et al. (2011).

*Line 126: the "and" after "data" seems out of place.*
*Line 141-142, "This tilt is consistent with equatorward flow through the channel". This statement is at odds with a generally poleward flow in the very nearshore band and in the SPC, which tends to bring waters from the southern bight, and is embedded in the Southern California Eddy. The predominantly equatorward California Current is much more offshore.*
This is seasonally and event dependent. The mean circulation is poleward as the Reviewer indicates. Todd et al. (2009) showed this during the HB06 (fall 2006) experiment, and Hickey et al. (2003) supports this. However, examining the spring data from Hickey's T1 mooring shows frequent reversals (toward downcoast), and Noble et al. (2002) show frequent reversals at mooring B 30m depth.

[Figure]

Figure R4: Nitrate versus temperature at SPOT (2000 to 2011). A linear relationship between nitrate and temperature is observed for temperatures cooler than 12.5°C. Between 12.5°C and 14°C nitrate goes to zero and remains very low. Based on this relationship, we use the 12.5°C isotherm as a proxy for higher nutrient waters that are within the nitricline.

*Line 170, "seasonal traits" clarify o rephrase.*
*Line 208-209, "we cannot distinguish between local and remote sources": this is at odds with the previous sentence, and with the general notion that the glider data allows a characterization of the spatial domain of the time series data. Please clarify.*

The glider captures the variability along the transect, but doesn't fully inform us about along coast transport and continuity of features along the coast. Two to three parallel glider lines perpendicular to the coast would provide a better sense of along coast transport but this unfortunately was beyond the scope of this project.  We have added text to clarify this.

*Line 214, "maintained an onshore-offshore gradient on average": please clarify the sentence.*
*Line 244, "highly offshore characteristics (Figure 6a)": this seems to refer to the wrong panel, please double check.*
*Line 249-250: this sentence seems to undermine the idea of connectivity as a material path connecting inshore and offshore.*
*Line 288, "bloom thickness": defined how?*
*Line 321, "decreased sedimentation": please clarify.*
*Line 331, "events": the term doesn't seem appropriate for the modes considered, especially deep chlorophyll maxima which seem the common "background" state for the SPC. Please clarify.*

*Figure 1. Some bin numbers could be useful, e.g. 10,20,58, since they are used later.*
We have added these.
*Figure 2. The mean mixed layer could be shown on this figure too for the 4 end- members.*

We have indicated MLD on the Temperature plot.

*Figure 3. It would be useful to have a measure of the distance from the Palos Verdes Peninsula together with the bin numbers in the x-axis. Also, it would be useful to add the location of SPOT in the panels. The caption should also specify the months of the observations, besides the years.*
We have made these changes.

*Figure 4. Are two panels really necessary? It seems that all information of panel b could be contained in panel a. Also, how were the ellipses defined?*
We have updated Figure 4 (see *above*) to include both the glider profiles and the ship-based profiles. Ellipse definition is now described in the Extended Methods.

*Figure 5 is a bit messy, and could go into the Supplement. It is not very successful in showing a clear seasonal progression or cycle, like the similar figures in Jacox, 2016 that presumably inspired it. To me the message is that the potential seasonal progression of upwelling is swamped by high frequency variability. (Or that perhaps the 2 PCA axes are not well positioned to highlight this progression.)*
We have edited Figure 5 to highlight the seasonal progression in PCA space and have moved the original figure in the Supplement (*see attached revised Figure 5*). The edited Figure 5 uses PCA space to track the seasonal progression of water mass profile types in the San Pedro Channel.

[Figure]

Figure 5: Seasonal progression of bins along the transcet moving from low PC1 values to elevated PC1 values. Two examples are shown: SPOT, and the average for bins 55-62. The average and standard deviation for April (blues) and June (greens) are shown.

*Figure 7: I wonder if the ratio between surface and total integrated cha may be a better variable to show here.*

We have edited this figure to show the relationship between the second principal coordinate axis, total integrated chlorophyll over 70 meters, and the integrated chlorophyll within the first optical depth (*see revised Figure 7 on page 8*).

*Supplementary Table S1: many of the thresholds and combinations behind the end-member definition seem somewhat arbitrary, and could be better justified, e.g. with a dedicated Supplement section.*

We have added an Extended Methods (*see attached*).

*Supplementary Figure S2: what are the vectors on the plot? Please explain in the caption.*

This is now explained in the Extended Methods (*see attached*).

*Supplementary Figure S3: please correct the units in the caption of panel a, from ug to mg/m3.*

*Supplementary Figure S4: the legend of the figure states "Satellite:Glider" mach-ups, but the labels and caption state "Glider-Satellite", please clarify.*

This has been fixed.

Table 1: Distribution of profile types as estimated by the structured PCA. The number of profiles that fall within with each end-member group (95% confidence interval) is given (N, %) for: all glider profiles (all glider), glider profiles from the SPOT bin (SPOT glider), all ship-based SPOT profiles (SPOT cruise), ship-based SPOT profiles from the months that the gliders were in the water (SPOT cruise March-July), all UpRISEE ship-based profiles (UpRISEE cruise), and UpRISEE profiles from the months that the gliders were in the water (UpRISEE cruise March-July).

| | All Glider Samples (March-July) | | SPOT Glider Samples (March-July) | | SPOT Cruise Samples (All Months) | | SPOT Cruise Samples (March-July) | | UpRISEE Cruise Samples (All Months) | | UpRISEE Cruise Samples (March-July) | |
|---|---|---|---|---|---|---|---|---|---|---|---|---|
| | N | % | N | % | N | % | N | % | N | % | N | % |
| Total Sample Size | 1606 | -- | 54 | -- | 64 | -- | 30 | -- | 21 | -- | 14 | -- |
| Cold High-Chlorophyll | 19 | 1.2% | 1 | 1.8% | 0 | 0.0% | 0 | 0.0% | 0 | 0.0% | 0 | 0.0% |
| Warm Sub-surface High Chlorophyll | 398 | 24.8% | 14 | 25.9% | 10 | 15.6% | 2 | 6.6% | 4 | 19.0% | 3 | 21.4% |
| Warm Low-Chlorophyll | 194 | 12.1% | 5 | 9.3% | 8 | 12.5% | 4 | 13.3% | 4 | 19.0% | 3 | 21.4% |
| Cold Low-Chlorophyll | 25 | 1.6% | 0 | 0.0% | 0 | 0.0% | 0 | 0.0% | 0 | 0.0% | 0 | 0.0% |
| Not Within 95% C.I. | 970 | 60.4% | 34 | 63.0% | 46 | 71.8% | 24 | 80.0% | 13 | 61.9% | 8 | 57.1% |

**References:**

Bialonski, S., Caron, D. A., Schloen, J., Feudel, U., Kantz, H., and Moorthi, S. D.: Contribution to the Themed Section: Scaling from individual lankton to marine ecosystems Phytoplankton dynamics in the Southern California Bight indicate a complex mixture of transport and biology, Journal of Plankton Research, 38, 1077-1091, 10.1093/plankt/fbv122, 2016.

Di Lorenzo, E.: Seasonal dynamics of the surface circulation in the Southern California Current System, Deep Sea Research Part II: Topical Studies in Oceanography, 50, 2371-2388, 2003.

Esaias, W. E., Abbott, M. R., Barton, I., Brown, O. B., Campbell, J. W., Carder, K. L., Clark, D. K., Evans, R. H., Hoge, F. E., Gordon, H. R., Balch, W. M., Letelier, R., and Minnett, P. J.: An overview of MODIS capabilities for ocean science observations, Ieee Transactions on Geoscience and Remote Sensing, 36, 1250-1265, 10.1109/36.701076, 1998.

Gordon, H. R., and McCluney, W. R.: Estimation of the Depth of Sunlight Penetration in the Sea for Remote Sensing, Applied Optics, 14, 413-416, 10.1364/ao.14.000413, 1975.

Haskell, W. Z., Prokopenko, M. G., Hammond, D. E., Stanley, R. H. R., Berelson, W. M., Baronas, J. J., Fleming, J. C., and Aluwihare, L.: An organic carbon budget for coastal Southern California determined by estimates of vertical nutrient flux, net community production and export, Deep-Sea Research Part I-Oceanographic Research Papers, 116, 49-76, 10.1016/j.dsr.2016.07.003, 2016.

Hickey, B. M.: Circulation over the Santa Monica-San Pedro Basin and Shelf, Progress in Oceanography, 30, 37-115, https://doi.org/10.1016/0079-6611(92)90009-O, 1992.

Hickey, B. M., Dobbins, E. L., and Allen, S. E.: Local and remote forcing of currents and temperature in the central Southern California Bight, Journal of Geophysical Research-Oceans, 108, 10.1029/2000jc00313, 2003.

Hoge, F. E., Wright, C. W., Lyon, P. E., Swift, R. N., and Yungel, J. K.: Satellite retrieval of the absorption coefficient of phytoplankton phycoerythrin pigment: theory and feasibility status, Applied Optics, 38, 7431-7441, 10.1364/ao.38.007431, 1999.

Jacox, M. G., Edwards, C. A., Kahru, M., Rudnick, D. L., and Kudela, R. M.: The potential for improving remote primary productivity estimates through subsurface chlorophyll and irradiance measurement, Deep-Sea Research Part Ii-Topical Studies in Oceanography, 112, 107-116, 10.1016/j.dsr2.2013.12.008, 2015.

Kirk, J. T.: Light and photosynthesis in aquatic ecosystems, Cambridge university press, 1994.

Lucas, A. J., Dupont, C. L., Tai, V., Largier, J. L., Palenik, B., and Franks, P. J. S.: The green ribbon: Multiscale physical control of phytoplankton productivity and community structure over a narrow continental shelf, Limnology and Oceanography, 56, 611-626, 10.4319/lo.2011.56.2.0611, 2011.

Mitarai, S., Siegel, D. A., Watson, J. R., Dong, C., and McWilliams, J. C.: Quantifying connectivity in the coastal ocean with application to the Southern California Bight, Journal of Geophysical Research-Oceans, 114, 10.1029/2008jc005166, 2009.

Montes-Hugo, M. A., Gould, R., Arnone, R., Ducklow, H., Carder, K., English, D., Schofield, O., and Kerfoot, J.: Beyond the first optical depth: fusing optical data from ocean color imagery and gliders. Montes-Hugo, M. A. (Ed.), 2009.

Morel, A., and Berthon, J. F.: SURFACE PIGMENTS, ALGAL BIOMASS PROFILES, AND POTENTIAL PRODUCTION OF THE EUPHOTIC LAYER - RELATIONSHIPS REINVESTIGATED IN VIEW OF REMOTE-SENSING APPLICATIONS, Limnology and Oceanography, 34, 1545-1562, 10.4319/lo.1989.34.8.1545, 1989.

Noble, M., Jones, B., Hamilton, P., Xu, J. P., Robertson, G., Rosenfeld, L., and Largier, J.: Cross-shelf transport into nearshore waters due to shoaling internal tides in San Pedro Bay, CA, Continental Shelf Research, 29, 1768-1785, 10.1016/j.csr.2009.04.008, 2009a.

Noble, M. A., Ryan, H. F., and Wiberg, P. L.: The dynamics of subtidal poleward flows over a narrow continental shelf, Palos Verdes, CA, Continental Shelf Research, 22, 923-944, 10.1016/s0278-4343(01)00112-1, 2002.

Noble, M. A., Rosenberger, K. J., Hamilton, P., and Xu, J. P.: Coastal ocean transport patterns in the central Southern California Bight, Earth Science in the Urban Ocean: the Southern California Continental Borderland, 454, 193-226, 10.1130/2009.2454(3.3), 2009b.

Sathyendranath, S., Prieur, L., and Morel, A.: A three-component model of ocean colour and its application to remote sensing of phytoplankton pigments in coastal waters, International Journal of Remote Sensing, 10, 1373-1394, 10.1080/01431168908903974, 1989.

Seegers, B. N., Birch, J. M., Marin, R., Scholin, C. A., Caron, D. A., Seubert, E. L., Howard, M. D. A., Robertson, G. L., and Jones, B. H.: Subsurface seeding of surface harmful algal blooms observed through the integration of autonomous gliders, moored environmental sample processors, and satellite remote sensing in southern California, Limnology and Oceanography, 60, 754-764, 10.1002/lno.10082, 2015.

Stramska, M., and Stramski, D.: Effects of a nonuniform vertical profile of chlorophyll concentration on remote-sensing reflectance of the ocean, Applied Optics, 44, 1735-1747, 10.1364/ao.44.001735, 2005.

Stukel, M. R., Song, H., Goericke, R., and Miller, A. J.: The role of subduction and gravitational sinking in particle export, carbon sequestration, and the remineralization length scale in the California Current Ecosystem, Limnology and Oceanography, 63, 363-383, 10.1002/lno.10636, 2018.

Todd, R. E., Rudnick, D. L., and Davis, R. E.: Monitoring the greater San Pedro Bay region using autonomous underwater gliders during fall of 2006, Journal of Geophysical Research-Oceans, 114, 10.1029/2008jc005086, 2009.

Wang, M. H., Son, S., and Shi, W.: Evaluation of MODIS SWIR and NIR-SWIR atmospheric correction algorithms using SeaBASS data, Remote Sensing of Environment, 113, 635-644, 10.1016/j.rse.2008.11.005, 2009.

**Expanded Methods:**

*Glider Data:* After all glider data had been gridded onto the cross-channel glider transect and partial glider profiles had been removed, secondary profile metrics were calculated for each profile. For each profile, we estimated the euphotic depth following the regionally validated method described in Jacox et al. (2015), which utilizes in-situ chlorophyll profiles and satellite daily PAR data. For this study, we used 9 kilometer satellite surface PAR data from MODIS Aqua for all light profile calculations. Euphotic depths during the 2013 and 2014 glider deployments were in good agreement with those collected from *in situ* PAR measurements during the concurrent Upwelling Regime In-Situ Ecosystem Efficiency study (UpRISEE) cruises at the SPOT site (Haskell et al., 2016).

Temperature profiles were used to calculate the mixed layer depth (MLD), mixed layer temperature (MLTemp), and the depth of the 12.5°C isotherm (z12p5). Chlorophyll a fluorescence profiles were used to calculate the depth of the chlorophyll maximum (zMaxChl), the maximum value of chlorophyll a along the profile (maxCHL), the integrated chlorophyll a within the top 70 meters (ChlInt70), and the ratio of integrated chlorophyll a within the top 20 meters versus the top 70 meters (ChlInt70Per20). Twenty meters was used to approximate the average mixed layer depth and surface thickness. Seventy meters was chosen as the maximum depth of chlorophyll integration as it included the full euphotic depth for 99% of the glider profiles from 2013 and 2014. In addition, we estimated from the ship-based SPOT time-series data (2003 - 2011) that on average PAR at 70m was 2.6% of the surface value, with a maximum of 4.5%. Ideally integrated chlorophyll would always be calculated for the entire euphotic zone, however here we were constrained to the upper 70 m due to the dive depth of the gliders.

*Principal Component Analysis (PCA):* All glider profiles from 2013 and 2014 were combined into a single PCA after normalization and standardization of the 7 profile characteristics described above (*Figure R3*). The PCA loadings (eigenvectors) are also plotted showing the loading (weighting) of each of the original variables onto the PCA axes.

The results of this PCA showed that most profiles were clustered together, with a small subset of cold, high chlorophyll (nominally surface bloom) profiles driving much of the separation on both PC1 and PC2 (*Figure R3*). Analysis of this un-structured PCA suggested that there were meaningful distinctions within the large cluster of profiles. Specifically, two end-members within this cluster were apparent: 1) a cool, deep MLD, low chlorophyll water column profile type and 2) a warm, shallow MLD, low surface chlorophyll water column profile type. Based on the un-structured PCA, we defined three 'end-member' water column profile types:
  *(1) cool, high chlorophyll (CHC)*
  *(2) cool, low chlorophyll (CLC)*
  *(3) warm, subsurface high chlorophyll (WSHC)*

These three water column profile types align with our understanding of oceanographic states of the region. Specifically, periodic upwelling along the coast brings cool, high nutrient, low chlorophyll waters to the surface. The early stages of this upwelling, before the initiation of a bloom, resulted in the CLC end-member. The cool, high chlorophyll (CHC) end-member typified a coastal surface phytoplankton bloom that classically follows coastal upwelling. Finally, warm profiles with subsurface high chlorophyll (WSHC) are classic profiles for relatively oligotrophic waters with stronger subsurface chlorophyll maxima.

In addition to the three end-members identified in the un-structured PCA, we identified a fourth, unique, end-member water profile type based on our examination of the glider dataset and our understanding of the oceanography of the San Pedro Channel. This fourth type represented the oligotrophic end-member and was termed:

       *(4) warm, low chlorophyll (WLC)*

This end-member typified relatively oligotrophic, warm, offshore waters with low chlorophyll throughout the upper water column that were being brought into the San Pedro Channel most likely as a result of the Southern California Eddy. All four of these end-member water column profile types were common within the glider curtain plots from the 2013 and 2014 glider deployments.

To identify these end-member profile types in an unbiased manner, we developed numerical criteria in order to quantitatively isolate the end-member profiles (Supplemental Table S1). For end-member types 1-3, we started with the profiles identified in the un-structured PCA and refined the criteria in order to isolate the most 'pure' examples of these water mass profile types. Using our criteria, we identified 54 'end-member profiles': 10 for type 1 (CHC), 12 for type 2 (CLC), 15 for type 3 (WSHC), and 17 for type 4 (WLC). For another region, where different end-member water profile types are present, separate numerical criteria would need to be developed in order to select representative end-member profiles for that dataset.

In order to differentiate profile characteristics more efficiently and to better represent the observed variability within the glider profile types, we conducted a second PCA which used only the 54 'end-member' profiles to define the PCA axes (hereafter referred to as the structured PCA). The remaining glider profiles were then projected onto these end-member defined PCA axes using the function proj within R software. Confidence intervals of 95% were calculated for each of the end-member groups based on PC1 and PC2 values for end-member profiles using the iso-contour of the Gaussian distribution after www.visiondummy.com/2014/04/draw-error-ellipse-representing-covariance-matrix/ and the function ggbiplot within R software. In brief, the magnitude of ellipse axes were determined by the variance within each end-member group, defined as the eigenvalues from the covariance matrix. The direction of the major axis was calculated from the eigenvector of the covariance matrix that corresponded to the largest eigenvalue.

The glider profiles were analyzed within the structured PCA space with specific focus on temporal and spatial changes, as well as how the profiles taken at the SPOT site related to and varied with profiles from the rest of the San Pedro Channel cross-section. The structured PCA resulted in clear separation of the 4 end-member profile types and allowed for better overall separation of the glider profiles. Specifically, the percent of explained variance for PC1 increased from 40.5% to 49.8%, and the percent of explained variance for PC2 increased from 21.5% to 32.7%. More importantly, the structured PCA allowed for more meaningful separation of the glider profiles into oceanographically relevant states allowing us to better understand and quantify the overall variance within the glider profile data. For example, movement along the first principal coordinate axis described changes in temperature-based characteristics while movement along the second principal coordinate axis described changes in chlorophyll-based characteristics. In addition, because the total variance associated with the PC1 and PC2 axes was similar, the distances in principal coordinate space could be used to define similarity between profiles.

***Comparison with time-series measurements***: To interpret the San Pedro Ocean Time-series data (monthly sampling) within the context of the variability identified in the high resolution glider dataset, we incorporated these ship-based measurements into our analysis (Table 1). Specifically, profile characteristics (*described above*) were calculated for 64 ship-based SPOT profiles from 2000-2011. 30 of these profiles fell between March and July, the time-period during which the gliders were in the water. In addition, 21 profiles were calculated from the UpRISEE ship-based cruises that occurred every two weeks during 2013 and 2014 (Haskell et al., 2016), with 14 profiles occurring between March and July. Though there was good coherence between temperature measurements across all three datasets, the chlorophyll fluorescence measurements from the 2000-2011 SPOT site cruises were considerably lower than the fluorescence measurements from both in-situ gliders and ship-based cruises from 2013 to 2014. We assumed this to be inter-instrument variation in fluorescence to chlorophyll ratio, rather than changes in in-situ chlorophyll concentration itself. To allow for projection onto the glider-derived structured PCA axes, the chlorophyll fluorescence data from 2000-2011 SPOT cruises were scaled so that the March through July mean chlorophyll fluorescence value was equal to the March through July mean chlorophyll fluorescence value from the 2013 to 2014 UpRISEE cruises.

The profile characteristics (MLD, MLTemp, z12p5, zMaxChl, maxCHL, chlInt70, and chlInt70Per20) for the 85 ship-based profiles were used to project these samples onto the end-member based PCA axes. Corresponding PC1 and PC2 values for all ship-based SPOT site profiles were then compared with glider profiles to assess interannual profile variability at the SPOT site (*Figure 4*).

**References:**

Haskell, W. Z., Prokopenko, M. G., Hammond, D. E., Stanley, R. H. R., Berelson, W. M., Baronas, J. J., Fleming, J. C., and Aluwihare, L.: An organic carbon budget for coastal Southern California determined by estimates of vertical nutrient flux, net community production and export, Deep-Sea Research Part I-Oceanographic Research Papers, 116, 49-76, 10.1016/j.dsr.2016.07.003, 2016.

Jacox, M. G., Edwards, C. A., Kahru, M., Rudnick, D. L., and Kudela, R. M.: The potential for improving remote primary productivity estimates through subsurface chlorophyll and irradiance measurement, Deep-Sea Research Part Ii-Topical Studies in Oceanography, 112, 107-116, 10.1016/j.dsr2.2013.12.008, 2015.

[Figure]

Figure R3: Un-structured Principal Component Analysis of glider profiles. The four end-member water column profile types and the SPOT glider profiles are highlighted. The black arrows denote the the loadings of each variable used in the PCA.

[Figure]

Figure 4: Structured Principal Component Analysis of glider and ship-based profiles. The four end-member water column profile types (Supplemental Figure S2) were used to create principal component axes. Physical variability was associated with PC1 (49.8% of total variance) and biological variability was associated with PC2 (32.7% of total variance). Panel A shows all data projected onto these axes including: all glider profiles collected in 2013 and 2014 (grey dots), glider profiles from the SPOT location (black diamonds), SPOT ship-based profiles (grey squares), and UpRISEE ship-based profiles (black squares).

---

## Author Comment (AC2) · 3 Apr 2018

*This paper uses high-frequency spatial and temporal glider data to quantify variability at the coastal San Pedro Ocean Time-series (SPOT) site in the San Pedro Channel (SPC) and provide insight into the underlying oceanographic dynamics for the site.*

*The glider data (a total of 1606 profiles) collected from March through July of 2013 and 2014 are used. This is a very rich data set and a detailed analysis is well justified for a publication. However, the manuscript in its current form is very difficult to read and follow. PCA is used to differentiate different profile types. It is confusing how the 54 end-member profiles are selected to define each of four dominant profile types, and then the remaining 1552 profiles are then projected onto the PC1 and PC2 coordinates. Maybe a more detailed description of the methodology is needed in the supplemental information.*

To address the overall concerns raised by the Reviewer, we have edited the text to clarify the methods, added additional analyses, and included a comparison between the glider data and ship-based SPOT measurements. To address the specific concern regarding the selection of the end-member profiles, we have added in an expanded methodology to clarify the way in which the main modes of variability (end-members) were identified and how the PCA analysis was conducted (*see Extended Methods attached at the end of this response*).

*Time series are mentioned as the motivation of this paper, although the SPOT data are not used in the analysis. Both weekly and monthly time scales are mentioned in the text, what is the time interval for the SPOT measurements? It is not true "most time- series are sampled ... approximately once per month. Many time series use mooring platforms collecting data every few minutes.*

We have clarified the text to highlight our goal of using high-resolution data-sets to provide context for time-series with low sampling frequency. We have also added a comparison between the glider profiles and both ship-based SPOT data from 2000-2011 and data from a set of cruises (UpRISEE) which occurred during the time of the glider deployments (2013-2014).

*p2, end of the 1st paragraph, "...at an individual site relative to a larger region may provide a path for leveraging numerous local time series sites in order to gain an understanding of larger scale oceanographic dynamics." What is the spatial scale for this "larger" region/scale? Maybe the SPOT time series can be used to quantify this spatial scale.*

We see this analysis as a proof of concept study of using high resolution glider data to determine whether coarse-resolution (monthly) sampling at a single point location is sufficient for capturing the variability within a larger (very dynamic) region. Here we use the SPOT time-series site as the point location and the San Pedro Channel as the larger region. However, based on previous work, we believe that the San Pedro Channel is in general representative of the larger Southern California Bight (e.g., Cullen and Eppley, 1981; Collins et al., 2011; Chow et al., 2013). The spatial scale for the 'larger' region will, however, be entirely site dependent and so we do not feel that our study is able to make a generalization on the larger spatial scale which can be

represented by time-series sites.  However, we do feel that our framework could be applied to other sites in order to answer this question.

*p2, 2nd paragraph, "cloud contamination" is not mentioned as the primary reason to have limited coverage.*

We have edited the text to include cloud contamination as a primary limitation of satellite measurements.

*"coastal and offshore processes", define "coastal" and "offshore"*
*It seems arbitrary to have the four dominant water column profile types: early upwelling, surface phytoplankton bloom, subsurface chlorophyll maximum, and offshore influence. Again define "offshore" here. Should the wind forcing be used?*

We have edited the text to clarify our terminology and choice of end-member water column profile types.  In addition, we have included an extended methods section providing additional detail on our methodology (*see Extended Methods attached at the end of this response*).

*p5, satellite data are mentioned, but should be used more to study the surface and subsurface linkage*

A large body of literature has commented on the relationship between satellite observable chlorophyll (within the first optical depth) and total integrated water column chlorophyll, as well as the need for increased *in situ* sampling to improve satellite chlorophyll and primary production algorithms due to mismatches and inconsistencies in modeling of the *in situ* chlorophyll profiles (e.g., Morel and Berthon, 1989; Stramska and Stramski, 2005; Sathyendranath et al., 1989; Montes-Hugo et al., 2009; Jacox et al., 2015; Seegers et al., 2015). Locally, while satellite surface chlorophyll estimates have been shown to aligned closely with *in situ* glider observations of nearshore surface blooms in the San Pedro Channel, subsurface chlorophyll layers farther offshore were undetected by satellite retrievals (Seegers et al., 2015). Here, we used our framework to identify which oceanographic states for the SPOT site may be most susceptible to satellite misinterpretation. We specifically avoided interpretation of the mismatch between the glider and satellite chlorophyll estimates for a number of reasons.  Primarily, we do not find this mismatch surprising given the differences in temporal and spatial scales of these two measurements. Specifically, the glider data were collected at ~0.5km resolution continuously over 24 hours through the deployment while the satellite measurements represent a single pass every 1 to 2 days in the afternoon and averaged over 1 km (Esaias et al., 1998). Furthermore, inaccuracies in the CDOM corrections and atmospheric corrections could also contribute to the observed mismatch (Esaias et al., 1998; Hoge et al., 1999; Wang et al., 2009). We do not feel that our dataset or analysis is the correct one to evaluate the robustness of the MODIS product for the San Pedro Channel.

We believe the important take-away from this part of our analysis is that the inherent bias in satellite data of only quantifying chlorophyll over the first optical depth is not a significant issue

for samples with high PC2 values and low overall biomass. However, it becomes an increasingly important issue for samples with high biomass and low PC2 values, which have a large percentage of integrated chlorophyll beneath the first optical depth. Specifically, the sub-surface chlorophyll profiles for these samples deviate significantly from the traditional relationship between chlorophyll within OD1 and total integrated chlorophyll. The structured PCA framework provides a metric for assessing the water mass types that may be most problematic for the satellite algorithms. Specifically, our analysis suggests that the satellite vs integrated chlorophyll mismatch may be particularly problematic for some cool, high chlorophyll water mass types (nominally coastal blooms). This suggests that increased *in situ* sampling may be needed when these water mass types are present in order to accurately constrain estimates of biomass distributions and primary production. We have edited the text and Figure 7 to clarify our incorporation of satellite data into our analyses.

[Figure]

Figure 7: Integrated chlorophyll within the first optical depth (OD1) versus total integrated chlorophyll over 70 m for all 2013 and 2014 glider profiles. Each profile is colored by its PC2 value. The location of the four end-member profiles types are also shown.

*p12, 5. Conclusion, end of the 1st paragraph, "...insensitive to coastal anthropogenic change...well positioned to identify a regional response to climate change." how do you derive such a conclusion?*

In the big picture, the SPOT time-series site is a coastal site. However, our analysis indicates that SPOT is more reflective of the offshore stratified environment rather than the nearcoastal upwelling environment where coastal discharge (outfalls and storm water) are a more significant factor. As such, we conclude that SPOT is more likely to be influenced by regional responses to climatic shifts than to local events (e.g. increased discharge into the port of Los Angeles). We have edited the text to clarify this point.

*Table 1, define "SPOT specific profiles", "SPOT samples", what is CI? C2*

We have expanded Table 1 to include an analysis of the ship-based SPOT measurements (*see attached revised Table 1*) and edited the caption to clarify the contents of the table.

*Figure 1, I understand the color represents bathymetry, why don't you state this in the caption?*

We have edited the caption.

*Figure 2, what is the arrows mean below the figure, PC1, PC2? what does the "n=" mean?*

The PC1 and PC2 arrows indicate the separation of the end-member profiles on the PCA axes. For example, subsurface chl max was associated with high PC1 values while early upwelling was associated with low PC1 values. N refers to the number of glider profiles used to define these end-member profiles. We have edited the caption and text to clarify this.

*Figure 3, is "box plot" a more standard term than "whisker plot", see https://en.wikipedia.org/wiki/Box_plot; define "bin"*

We have edited the caption.

*Supplemental Figure S2, define "ideal profiles"*

We have included an Extended Methods sections describing how the end-member profiles were defined. We have replaced 'ideal profiles' with 'end-member' profiles throughout the text.

Table 1: Distribution of profile types as estimated by the structured PCA. The number of profiles that fall within with each end-member group (95% confidence interval) is given (N, %) for: all glider profiles (all glider), glider profiles from the SPOT bin (SPOT glider), all ship-based SPOT profiles (SPOT cruise), ship-based SPOT profiles from the months that the gliders were in the water (SPOT cruise March-July), all UpRISEE ship-based profiles (UpRISEE cruise), and UpRISEE profiles from the months that the gliders were in the water (UpRISEE cruise March-July).

| | All Glider Samples (March-July) | | SPOT Glider Samples (March-July) | | SPOT Cruise Samples (All Months) | | SPOT Cruise Samples (March-July) | | UpRISEE Cruise Samples (All Months) | | UpRISEE Cruise Samples (March-July) | |
|---|---|---|---|---|---|---|---|---|---|---|---|---|
| | N | % | N | % | N | % | N | % | N | % | N | % |
| Total Sample Size | 1606 | -- | 54 | -- | 64 | -- | 30 | -- | 21 | -- | 14 | -- |
| Cold High-Chlorophyll | 19 | 1.2% | 1 | 1.8% | 0 | 0.0% | 0 | 0.0% | 0 | 0.0% | 0 | 0.0% |
| Warm Sub-surface High Chlorophyll | 398 | 24.8% | 14 | 25.9% | 10 | 15.6% | 2 | 6.6% | 4 | 19.0% | 3 | 21.4% |
| Warm Low-Chlorophyll | 194 | 12.1% | 5 | 9.3% | 8 | 12.5% | 4 | 13.3% | 4 | 19.0% | 3 | 21.4% |
| Cold Low-Chlorophyll | 25 | 1.6% | 0 | 0.0% | 0 | 0.0% | 0 | 0.0% | 0 | 0.0% | 0 | 0.0% |
| Not Within 95% C.I. | 970 | 60.4% | 34 | 63.0% | 46 | 71.8% | 24 | 80.0% | 13 | 61.9% | 8 | 57.1% |

**References:**

Chow, C. E. T., Sachdeva, R., Cram, J. A., Steele, J. A., Needham, D. M., Patel, A., Parada, A. E., and Fuhrman, J. A.: Temporal variability and coherence of euphotic zone bacterial communities over a decade in the Southern California Bight, Isme Journal, 7, 2259-2273, 10.1038/ismej.2013.122, 2013.

Collins, L. E., Berelson, W., Hammond, D. E., Knapp, A., Schwartz, R., and Capone, D.: Particle fluxes in San Pedro Basin, California: A four-year record of sedimentation and physical forcing, Deep-Sea Research Part I-Oceanographic Research Papers, 58, 898-914, 10.1016/j.dsr.2011.06.008, 2011.

Cullen, J. J., and Eppley, R. W.: Chlorophyll maximum layers of the Southern-California Bight and possible mechanisms of their formation and maintenance, Oceanologica Acta, 4, 23-32, 1981.

Esaias, W. E., Abbott, M. R., Barton, I., Brown, O. B., Campbell, J. W., Carder, K. L., Clark, D. K., Evans, R. H., Hoge, F. E., Gordon, H. R., Balch, W. M., Letelier, R., and Minnett, P. J.: An overview of MODIS capabilities for ocean science observations, Ieee Transactions on Geoscience and Remote Sensing, 36, 1250-1265, 10.1109/36.701076, 1998.

Hoge, F. E., Wright, C. W., Lyon, P. E., Swift, R. N., and Yungel, J. K.: Satellite retrieval of the absorption coefficient of phytoplankton phycoerythrin pigment: theory and feasibility status, Applied Optics, 38, 7431-7441, 10.1364/ao.38.007431, 1999.

Jacox, M. G., Edwards, C. A., Kahru, M., Rudnick, D. L., and Kudela, R. M.: The potential for improving remote primary productivity estimates through subsurface chlorophyll and irradiance measurement, Deep-Sea Research Part Ii-Topical Studies in Oceanography, 112, 107-116, 10.1016/j.dsr2.2013.12.008, 2015.

Montes-Hugo, M. A., Gould, R., Arnone, R., Ducklow, H., Carder, K., English, D., Schofield, O., and Kerfoot, J.: Beyond the first optical depth: fusing optical data from ocean color imagery and gliders. Montes-Hugo, M. A. (Ed.), 2009.

Morel, A., and Berthon, J. F.: SURFACE PIGMENTS, ALGAL BIOMASS PROFILES, AND POTENTIAL PRODUCTION OF THE EUPHOTIC LAYER - RELATIONSHIPS REINVESTIGATED IN VIEW OF REMOTE-SENSING APPLICATIONS, Limnology and Oceanography, 34, 1545-1562, 10.4319/lo.1989.34.8.1545, 1989.

Sathyendranath, S., Prieur, L., and Morel, A.: A three-component model of ocean colour and its application to remote sensing of phytoplankton pigments in coastal waters, International Journal of Remote Sensing, 10, 1373-1394, 10.1080/01431168908903974, 1989.

Seegers, B. N., Birch, J. M., Marin, R., Scholin, C. A., Caron, D. A., Seubert, E. L., Howard, M. D. A., Robertson, G. L., and Jones, B. H.: Subsurface seeding of surface harmful algal blooms observed through the integration of autonomous gliders, moored environmental sample processors,

and satellite remote sensing in southern California, Limnology and Oceanography, 60, 754-764, 10.1002/lno.10082, 2015.

Stramska, M., and Stramski, D.: Effects of a nonuniform vertical profile of chlorophyll concentration on remote-sensing reflectance of the ocean, Applied Optics, 44, 1735-1747, 10.1364/ao.44.001735, 2005.

Wang, M. H., Son, S., and Shi, W.: Evaluation of MODIS SWIR and NIR-SWIR atmospheric correction algorithms using SeaBASS data, Remote Sensing of Environment, 113, 635-644, 10.1016/j.rse.2008.11.005, 2009.

**Expanded Methods:**

*Glider Data:* After all glider data had been gridded onto the cross-channel glider transect and partial glider profiles had been removed, secondary profile metrics were calculated for each profile. For each profile, we estimated the euphotic depth following the regionally validated method described in Jacox et al. (2015), which utilizes in-situ chlorophyll profiles and satellite daily PAR data. For this study, we used 9 kilometer satellite surface PAR data from MODIS Aqua for all light profile calculations. Euphotic depths during the 2013 and 2014 glider deployments were in good agreement with those collected from *in situ* PAR measurements during the concurrent Upwelling Regime In-Situ Ecosystem Efficiency study (UpRISEE) cruises at the SPOT site (Haskell et al., 2016).

Temperature profiles were used to calculate the mixed layer depth (MLD), mixed layer temperature (MLTemp), and the depth of the 12.5˚C isotherm (z12p5). Chlorophyll a fluorescence profiles were used to calculate the depth of the chlorophyll maximum (zMaxChl), the maximum value of chlorophyll a along the profile (maxCHL), the integrated chlorophyll a within the top 70 meters (ChlInt70), and the ratio of integrated chlorophyll a within the top 20 meters versus the top 70 meters (ChlInt70Per20). Twenty meters was used to approximate the average mixed layer depth and surface thickness. Seventy meters was chosen as the maximum depth of chlorophyll integration as it included the full euphotic depth for 99% of the glider profiles from 2013 and 2014. In addition, we estimated from the ship-based SPOT time-series data (2003 - 2011) that on average PAR at 70m was 2.6% of the surface value, with a maximum of 4.5%. Ideally integrated chlorophyll would always be calculated for the entire euphotic zone, however here we were constrained to the upper 70 m due to the dive depth of the gliders.

*Principal Component Analysis (PCA):* All glider profiles from 2013 and 2014 were combined into a single PCA after normalization and standardization of the 7 profile characteristics described above (*Figure R3*). The PCA loadings (eigenvectors) are also plotted showing the loading (weighting) of each of the original variables onto the PCA axes.

The results of this PCA showed that most profiles were clustered together, with a small subset of cold, high chlorophyll (nominally surface bloom) profiles driving much of the separation on both PC1 and PC2 (*Figure R3*). Analysis of this un-structured PCA suggested that there were meaningful distinctions within the large cluster of profiles. Specifically, two end-members within this cluster were apparent: 1) a cool, deep MLD, low chlorophyll water column profile type and 2) a warm, shallow MLD, low surface chlorophyll water column profile type. Based on the un-structured PCA, we defined three 'end-member' water column profile types:

> *(1) cool, high chlorophyll (CHC)*
> *(2) cool, low chlorophyll (CLC)*
> *(3) warm, subsurface high chlorophyll (WSHC)*

These three water column profile types align with our understanding of oceanographic states of the region. Specifically, periodic upwelling along the coast brings cool, high nutrient, low chlorophyll waters to the surface. The early stages of this upwelling, before the initiation of a bloom, resulted in the CLC end-member. The cool, high chlorophyll (CHC) end-member typified a coastal surface phytoplankton bloom that classically follows coastal upwelling. Finally, warm profiles with subsurface high chlorophyll (WSHC) are classic profiles for relatively oligotrophic waters with stronger subsurface chlorophyll maxima.

In addition to the three end-members identified in the un-structured PCA, we identified a fourth, unique, end-member water profile type based on our examination of the glider dataset and our understanding of the oceanography of the San Pedro Channel. This fourth type represented the oligotrophic end-member and was termed:

*(4) warm, low chlorophyll (WLC)*

This end-member typified relatively oligotrophic, warm, offshore waters with low chlorophyll throughout the upper water column that were being brought into the San Pedro Channel most likely as a result of the Southern California Eddy. All four of these end-member water column profile types were common within the glider curtain plots from the 2013 and 2014 glider deployments.

To identify these end-member profile types in an unbiased manner, we developed numerical criteria in order to quantitatively isolate the end-member profiles (Supplemental Table S1). For end-member types 1-3, we started with the profiles identified in the un-structured PCA and refined the criteria in order to isolate the most 'pure' examples of these water mass profile types. Using our criteria, we identified 54 'end-member profiles': 10 for type 1 (CHC), 12 for type 2 (CLC), 15 for type 3 (WSHC), and 17 for type 4 (WLC). For another region, where different end-member water profile types are present, separate numerical criteria would need to be developed in order to select representative end-member profiles for that dataset.

In order to differentiate profile characteristics more efficiently and to better represent the observed variability within the glider profile types, we conducted a second PCA which used only the 54 'end-member' profiles to define the PCA axes (hereafter referred to as the structured PCA). The remaining glider profiles were then projected onto these end-member defined PCA axes using the function proj within R software. Confidence intervals of 95% were calculated for each of the end-member groups based on PC1 and PC2 values for end-member profiles using the iso-contour of the Gaussian distribution after www.visiondummy.com/2014/04/draw-error-ellipse-representing-covariance-matrix/ and the function ggbiplot within R software. In brief, the magnitude of ellipse axes were determined by the variance within each end-member group, defined as the eigenvalues from the covariance matrix. The direction of the major axis was calculated from the eigenvector of the covariance matrix that corresponded to the largest eigenvalue.

The glider profiles were analyzed within the structured PCA space with specific focus on temporal and spatial changes, as well as how the profiles taken at the SPOT site related to and varied with profiles from the rest of the San Pedro Channel cross-section. The structured PCA resulted in clear separation of the 4 end-member profile types and allowed for better overall separation of the glider profiles. Specifically, the percent of explained variance for PC1 increased from 40.5% to 49.8%, and the percent of explained variance for PC2 increased from 21.5% to 32.7%. More importantly, the structured PCA allowed for more meaningful separation of the glider profiles into oceanographically relevant states allowing us to better understand and quantify the overall variance within the glider profile data. For example, movement along the first principal coordinate axis described changes in temperature-based characteristics while movement along the second principal coordinate axis described changes in chlorophyll-based characteristics. In addition, because the total variance associated with the PC1 and PC2 axes was similar, the distances in principal coordinate space could be used to define similarity between profiles.

***Comparison with time-series measurements***: To interpret the San Pedro Ocean Time-series data (monthly sampling) within the context of the variability identified in the high resolution glider dataset, we incorporated these ship-based measurements into our analysis (Table 1). Specifically, profile characteristics (*described above*) were calculated for 64 ship-based SPOT profiles from 2000-2011. 30 of these profiles fell between March and July, the time-period during which the gliders were in the water. In addition, 21 profiles were calculated from the UpRISEE ship-based cruises that occurred every two weeks during 2013 and 2014 (Haskell et al., 2016), with 14 profiles occurring between March and July. Though there was good coherence between temperature measurements across all three datasets, the chlorophyll fluorescence measurements from the 2000-2011 SPOT site cruises were considerably lower than the fluorescence measurements from both in-situ gliders and ship-based cruises from 2013 to 2014. We assumed this to be inter-instrument variation in fluorescence to chlorophyll ratio, rather than changes in in-situ chlorophyll concentration itself. To allow for projection onto the glider-derived structured PCA axes, the chlorophyll fluorescence data from 2000-2011 SPOT cruises were scaled so that the March through July mean chlorophyll fluorescence value was equal to the March through July mean chlorophyll fluorescence value from the 2013 to 2014 UpRISEE cruises.

The profile characteristics (MLD, MLTemp, z12p5, zMaxChl, maxCHL, chlInt70, and chlInt70Per20) for the 85 ship-based profiles were used to project these samples onto the end-member based PCA axes. Corresponding PC1 and PC2 values for all ship-based SPOT site profiles were then compared with glider profiles to assess interannual profile variability at the SPOT site (*Figure 4*).

**References:**

Haskell, W. Z., Prokopenko, M. G., Hammond, D. E., Stanley, R. H. R., Berelson, W. M., Baronas, J. J., Fleming, J. C., and Aluwihare, L.: An organic carbon budget for coastal Southern California determined by estimates of vertical nutrient flux, net community production and export, Deep-Sea Research Part I-Oceanographic Research Papers, 116, 49-76, 10.1016/j.dsr.2016.07.003, 2016.

Jacox, M. G., Edwards, C. A., Kahru, M., Rudnick, D. L., and Kudela, R. M.: The potential for improving remote primary productivity estimates through subsurface chlorophyll and irradiance measurement, Deep-Sea Research Part Ii-Topical Studies in Oceanography, 112, 107-116, 10.1016/j.dsr2.2013.12.008, 2015.

[Figure]

Figure R3: Un-structured Principal Component Analysis of glider profiles. The four end-member water column profile types and the SPOT glider profiles are highlighted. The black arrows denote the the loadings of each variable used in the PCA.

[Figure]

Figure 4: Structured Principal Component Analysis of glider and ship-based profiles. The four end-member water column profile types (Supplemental Figure S2) were used to create principal component axes. Physical variability was associated with PC1 (49.8% of total variance) and biological variability was associated with PC2 (32.7% of total variance). Panel A shows all data projected onto these axes including: all glider profiles collected in 2013 and 2014 (grey dots), glider profiles from the SPOT location (black diamonds), SPOT ship-based profiles (grey squares), and UpRISEE ship-based profiles (black squares).

---

## Author Response (AR2)

*This is my second review of the manuscript by Teel and Coauthors. As stated in my initial review, the Authors describe a nice new dataset of glider observations that they then use to investigate the representativeness of the SPOT time-series dataset. My main criticisms were: the lack of use of actual SPOT data, a confusing description of the PCA end member-definition, and a weak comparison with satellite-based chlorophyll observations. I am pleased to see that the Authors have taken these and another Reviewer's comments seriously, and addressed them in revision. I appreciate the reorganization of the old figures, and a few new ones, and I find the revised manuscript clearer and easier to understand. The conclusions appear to be supported by the data, and there are a number of insights that should make the study valuable for the observational and modeling communities. Likewise, the glider observations will likely prove useful for anyone interested in the oceanography of the Southern California Bight, and in particular in the rich, long running SPOT dataset. I have a few minor comments that the Authors could further address; besides that, I am overall supportive of publication.*

Thank you for your constructive comments. We have addressed the final points in the revised manuscript. Please see details below.

*I would still carefully review the presentation of the two-step approach in the PCA analysis, to make sure it comes across as clear for the first-time reader. For example, in Section 2.4, the sentence starting in line 144 could further clarify the reason for a second PCA, the rationale for the 54 profiles selected, etc. Alternatively a reference could be added pointing to section 3.2, where more details are provided. Section 3.2 as well could benefit for an opening summary sentence, and could be clearly broken into two parts, one for the "original" PCA and the second for the "structured" PCA.*

To avoid repetition and to keep the manuscript as short as possible, we have decided to keep the full rationale for the second PCA and the end-member profile selection in the Results (*section 3.2*). However, we have added a brief motivation for the structured PCA to section 2.4 and refer the reader to section 3.2 on line 146. To help clarify the PCA analysis for the reader, we have also added an introductory paragraph to section 3.2.

*Figure 8, in particular panel (a) could be clarified. At first, I was confused by the number of dots of different colors, and it took me a while to figure out that longer re-sampling times increase the number of "sub-datasets" that can be extracted from the full timeseries, thus producing multiple dots. This could be explained in the caption. The Authors could also clarify in a sentence of two of text why the mean, S.D., and distance in PCA space are good measures of the ability of a sampling frequency to capture real variability. This is not completely obvious, given how abstract the PCA space is. I wonder if there is a better way to think about the impact of sub-sampling. For example, would a less frequent sampling strategy be able to recover the full end-members of the PCA (which include more rare, extreme states), rather than the mean state?*

We have added the number of sub-datasets per sampling scheme to the figure caption to help clarify this figure and added a sentence to the main text highlight that the number of sub-datasets varies as a results of sub-sampling of a finite number of data points. We have also included text describing why mean and standard deviation in PCA space are good metrics for analyzing the datasets.

Here we are analyzing the ability of sampling frequency to capture the mean state of the site. For this criteria, more frequent sampling is better. With more data points, the extreme states

have a smaller weight on the mean and so have less of an influence. However, these extreme states are still within the high-frequency dataset (e.g. the glider dataset). If the objective is to analyze the extreme states, then profiles need to be analyzed independently (e.g., as we have done with the PCA) not as aggregates.

*Abstract, line 22: I would change "was dominated by" with "could be described by a combination of" (or similar). Since most profiles are a combination of end members, none of the end members truly "dominates".*
   We have made this change.

*Line 207: "meaningful distinction" seems vague; also what kind of "analysis"? Please be more specific.*
   We have clarified this statement.

*Lines 222-227. I am still not convinced about the WSHC not being a true end member. After all, it is a real extreme in PCA space, and has warmer temperatures everywhere compared to other end members. Also, the deep chl maximum at ~30 m of WLC seems quite shallower than where I would expect a subtropical DCM (40-60m offshore the SCB). Finally, to the mechanisms listed in lines 224-227 for the DCM of the WSHC, I would add elevated NPP at depth along waters with high subducted NO3 (that is, nutrients, rather than just phytoplankton are subducted, stimulating growth at depth).*
   We have edited the description of the WSHC end-member and removed the disclaimer that it might not be a 'true end-member'. We have also added the comment that advection of upwelled nitrate could also lead to the DCM.

*Lines 342, 344: "correlation" rather than "relationship"?*
   We have made this change.

*Line 388: "Significant" has a statistical meaning that doesn't seem to apply to the eample of year 2001, which doesn't seem statistically different from the mean or the other years. Maybe use "Larger", or "Stronger" instead?*
   We have made this change.

*Please re-read carefully to fix some typos.*
   We have done a close read-through to catch typos.

[revised manuscript text omitted]